# SIV-specific neutralizing antibody induction following selection of a PI3K drive-attenuated *nef* variant

**Hiroyuki Yamamoto[1,2,3]\*, Tetsuro Matano[1,3,4]\***

[1]AIDS Research Center, National Institute of Infectious Diseases, Tokyo, Japan; [2]Department of Biomedicine, University Hospital Basel, Basel, Switzerland; [3]Joint Research Center for Human Retrovirus Infection, Kumamoto University, Kumamoto, Japan; [4]The Institute of Medical Science, The University of Tokyo, Tokyo, Japan

## eLife Assessment

Yamamoto and Matano provide **convincing** evidence that a G63E/R CD8+ T-cell escape mutation in the accessory viral protein Nef promote the induction of neutralizing antibody (nAb) responses in rhesus macaques infected with SIVmac239, which is usually largely resistant to neutralization. Functional analyses support that this mutation specifically impairs Nef's ability to stimulate PI3K/Akt/mTORC2 signaling. This **important** study suggests that the accessory viral protein Nef impairs B cell function and effective humoral immune responses and is of interest for researchers and physicians interested in HIV/AIDS and vaccine development.

**Abstract** HIV and simian immunodeficiency virus (SIV) infections are known for impaired neutralizing antibody (NAb) responses. While sequential virus–host B cell interaction appears to be basally required for NAb induction, driver molecular signatures predisposing to NAb induction still remain largely unknown. Here we describe SIV-specific NAb induction following a virus–host interplay decreasing aberrant viral drive of phosphoinositide 3-kinase (PI3K). Screening of seventy difficult-to-neutralize SIV$_{mac239}$-infected macaques found nine NAb-inducing animals, with seven selecting for a specific CD8$^+$ T-cell escape mutation in viral *nef* before NAb induction. This Nef-G63E mutation reduced excess Nef interaction-mediated drive of B-cell maturation-limiting PI3K/mammalian target of rapamycin complex 2 (mTORC2). In vivo imaging cytometry depicted preferential Nef perturbation of cognate Envelope-specific B cells, suggestive of polarized contact-dependent Nef transfer and corroborating cognate B-cell maturation post-mutant selection up to NAb induction. Results collectively exemplify a NAb induction pattern extrinsically reciprocal to human PI3K gain-of-function antibody-dysregulating disease and indicate that harnessing the PI3K/mTORC2 axis may facilitate NAb induction against difficult-to-neutralize viruses including HIV/SIV.

## Introduction

Virus-specific neutralizing antibody (NAb) responses by B cells are induced by an intricate cooperation of adaptive immune cells (***Kumar et al., 2010***; ***Shulman et al., 2014***; ***Gitlin et al., 2014***; ***Wang et al., 2014***) and often play a central role in clearance of acute viral infections (***Junt et al., 2007***). In contrast, persistence-prone viruses such as human immunodeficiency virus type 1 (HIV-1), simian immunodeficiency virus (SIV), and lymphocytic choriomeningitis virus (LCMV) variously equip themselves with B cell/antibody-inhibitory countermeasures (***Moir et al., 2001***; ***Mattapallil et al., 2005***; ***Sommerstein et al., 2015***; ***Sammicheli et al., 2016***; ***Fallet et al., 2016***; ***Mason et al., 2016***), impairing NAb

**\*For correspondence:**
h-yamato@niid.go.jp (HY);
tmatano@niid.go.jp (TM)

**Competing interest:** The authors declare that no competing interests exist.

induction (*Levesque et al., 2009*; *Mikell et al., 2011*). These viruses successfully suppress elicitation of potent NAb responses, especially in acute infection (*Hunziker et al., 2003*; *Tomaras et al., 2008*), and establish viral persistence, posing considerable challenges for developing protective strategies. In particular, HIV and SIV establish early a large body of infection in vivo from early on with a distinct host genome-integrating retroviral life cycle (*Whitney et al., 2014*). In addition, these lentiviruses are unique in launching a matrix of host immune-perturbing interactions mainly by their six remarkably pleiotropic accessory viral proteins, which optimally fuels viral pathogenesis (*Kirchhoff et al., 1995*; *Sheehy et al., 2002*; *Harris et al., 2003*; *Schindler et al., 2006*; *Neil et al., 2008*; *Zhang et al., 2009*; *Laguette et al., 2011*; *Yamada et al., 2018*; *Joas et al., 2018*; *Langer et al., 2019*; *Yan et al., 2019*; *Joas et al., 2020*; *Khan et al., 2020*; *Volcic et al., 2020*; *Reuschl et al., 2022*). A body of evidence has depicted this in the last decades, whereas its entity, including focal mechanisms of humoral immune perturbation in HIV/SIV infection, remains elusive to date.

Adverse virus–host interactions in HIV/SIV infection lead to a detrimental consequence of the absence of acute-phase endogenous NAb responses. Contrasting this, we and others have previously described in in vivo experimental models that passive NAb infusion in the acute phase can trigger an endogenous T-cell synergism, resulting in robust control of SIV and chimeric SHIV (simian/human immunodeficiency virus) (*Haigwood et al., 1996*; *Yamamoto et al., 2007*; *Ng et al., 2010*; *Iseda et al., 2016*; *Nishimura et al., 2017*). This indicates that virus-specific NAbs not only confer sterile protection but also can evoke T-cell-mediated non-sterile viral control, suggesting the importance of endogenous NAb responses supported by humoral-cellular response synergisms, during an optimal time frame. Therefore, identifying the mechanisms driving NAb induction against such viruses is an important step to eventually design NAb-based HIV control strategies.

One approach that can provide important insights into this goal is the analysis of in vivo immunological events linked with NAb induction against difficult-to-neutralize SIVs in a non-human primate model. Various in vivo signatures of HIV-specific NAb induction, such as antibody-NAb coevolution (*Moore et al., 2012*), autoimmune-driven induction (*Moody et al., 2016*), and natural killer cell-related host polymorphisms (*Bradley et al., 2018*), have been reported to date. The broad range of contributing factors collectively, and interestingly, indicates that pathways to NAb induction against difficult-to-neutralize viruses including HIV/SIV are redundant, and may potentially involve as-yet-unknown mechanisms driving NAb induction. For example, the neutralization resistance of certain SIV strains does not appear to be explained by any of the aforementioned, posing SIV models as attractive tools to analyze NAb induction mechanisms.

In the present study, we examined virus-specific antibody responses in rhesus macaques infected with a highly difficult-to-neutralize SIV strain, SIV$_{mac239}$. This virus is pathogenic in rhesus macaques causing simian AIDS across a broad range of geographical origin of macaques (*Cumont et al., 2008*). Macaques infected with SIV$_{mac239}$ show persistent viremia and generally lack NAb responses throughout infection (*Kestler et al., 1991*; *Nomura et al., 2012*). In this study, a large-scale screening of SIV$_{mac239}$-infected Burmese rhesus macaques for up to 100 weeks identified a subgroup inducing NAbs in the chronic phase. Interestingly, before NAb induction, these animals commonly selected for a specific CD8$^+$ cytotoxic T lymphocyte (CTL) escape mutation in the viral Nef-coding gene. Compared with wild-type (WT) Nef, this mutant Nef manifested a decrease in aberrant interaction with phosphoinositide 3-kinase (PI3K)/mammalian target of rapamycin complex 2 (mTORC2), resulting in decreased downstream hyperactivation of the canonical B-cell negative regulator Akt (*Omori et al., 2006*; *Limon et al., 2014*). Machine learning-assisted imaging cytometry revealed that Nef preferentially targets Env-specific B cells in vivo. Furthermore, the NAb induction was linked with sustained Env-specific B-cell responses after or during the mutant Nef selection. Thus, NAb induction in SIV$_{mac239}$-infected hosts conceivably involves a functional boosting of B cells that is phenotypically reciprocal to a recently found human PI3K gain-of-function and antibody-dysregulating inborn error of immunity (IEI), activated PI3 kinase delta syndrome (APDS) (*Angulo et al., 2013*; *Lucas et al., 2014*). Our results suggest that intervening PI3K/mTORC2 signaling can potentially result in harnessing NAb induction against difficult-to-neutralize viruses.

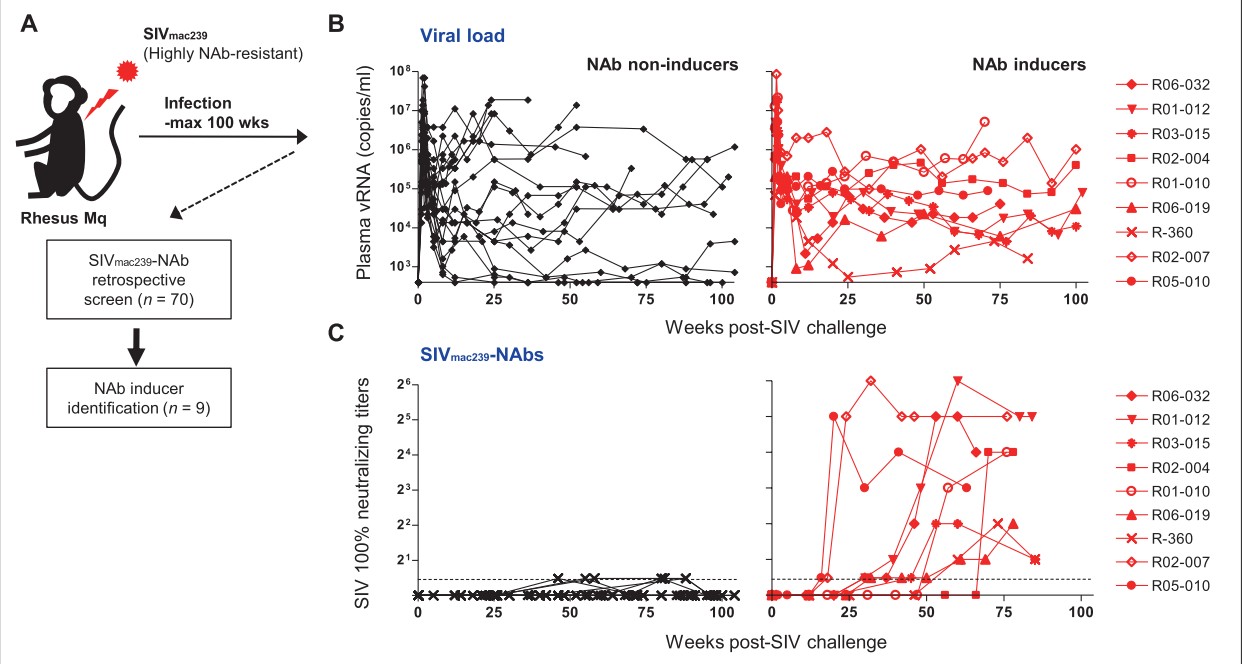

**Figure 1.** Neutralizing antibody (NAb) induction against NAb-resistant SIV$_{mac239}$. (**A**) Study design. (**B**) Plasma viral loads (simian immunodeficiency virus [SIV] *gag* RNA copies/ml plasma) in NAb non-inducers (left) and inducers (right). (**C**) Plasma SIV$_{mac239}$ 100% neutralizing end point titers by 10 TCID$_{50}$ killing assay on MT4-R5 cells. Points on the dotted line show marginally NAb-positive results (<1:2). In some animals, titers were comparable with our reported results using MT4 cells.

The online version of this article includes the following source data and figure supplement(s) for figure 1:

**Figure supplement 1.** Profile of SIV$_{mac239}$-NAb non-inducer subgroup cohort.

**Figure supplement 1—source data 1.** Source table for *Figure 1—figure supplement 1*.

**Figure supplement 2.** IgG class-switched neutralizing antibodies (NAbs) in the inducers.

**Figure supplement 2—source data 1.** Source table for *Figure 1—figure supplement 2*.

**Figure supplement 3.** Anti-SIV binding IgG profiles in neutralizing antibody (NAb) inducers and non-inducers.

**Figure supplement 3—source data 1.** PDF file containing original western blots for *Figure 1—figure supplement 3*, labeling the corresponding viral protein bands and IDs.

**Figure supplement 3—source data 2.** Original file for western blot analysis displayed in *Figure 1—figure supplement 3*.

**Figure supplement 4.** Env sequence variation pattern is not a major characteristic of neutralizing antibody (NAb) induction.

## Results

### Identification of macaques inducing SIV$_{mac239}$-neutralizing antibodies

We performed a retrospective antibody profile screening in rhesus macaques infected with NAb-resistant SIV$_{mac239}$ (n = 70) (*Figure 1A*) and identified a group of animals inducing anti-SIV$_{mac239}$ NAb responses (n = 9), which were subjected to characterization. These NAb inducers showed persistent viremia with no significant difference in viral loads compared with a subgroup of NAb non-inducers (n = 19) that were previously profile-clarified, naïve, and major histocompatibility complex class I (MHC-I) haplotype-balanced (used for comparison hereafter) (*Figure 1B*, *Figure 1—figure supplement 1*). Plasma SIV$_{mac239}$-NAb titers measured by 10 TCID$_{50}$ SIV$_{mac239}$ virus-killing assay showed an average maximum titer of 1:16, being induced at an average of 48 weeks post-infection (p.i.) (*Figure 1C*). Two of the nine NAb inducers showed detectable NAb responses by 24 weeks p.i., while the remaining seven induced NAbs after 30 weeks p.i. Anti-SIV$_{mac239}$ neutralizing activity was confirmed in immunoglobulin G (IgG) purified from plasma of these NAb inducers (*Figure 1—figure supplement 2*). SIV Env-binding IgGs developed from early infection in both NAb inducers and non-rapid-progressing NAb non-inducers, the latter differing from rapid progressors known to manifest serological failure (*Hirsch et al., 2004*; *Nakane et al., 2013*; *Figure 1—figure supplement 3A*). Titers of Env-binding IgG were not higher but rather lower at year 1 p.i. in the NAb

inducers (*Figure 1—figure supplement 3B and C*). This was consistent with other reports (*Havenar-Daughton et al., 2016*) and differing with gross B-cell enhancement in Nef-deleted SIV infection with enhanced anti-SIV binding antibody titers accompanying marginal neutralization (*Adnan et al., 2016*). NAb inducers and non-inducers showed similar patterns of variations in viral *env* sequences (*Burns et al., 1993*), mainly in variable regions 1, 2 and 4 (V1, V2, and V4) (*Figure 1—figure supplement 4*).

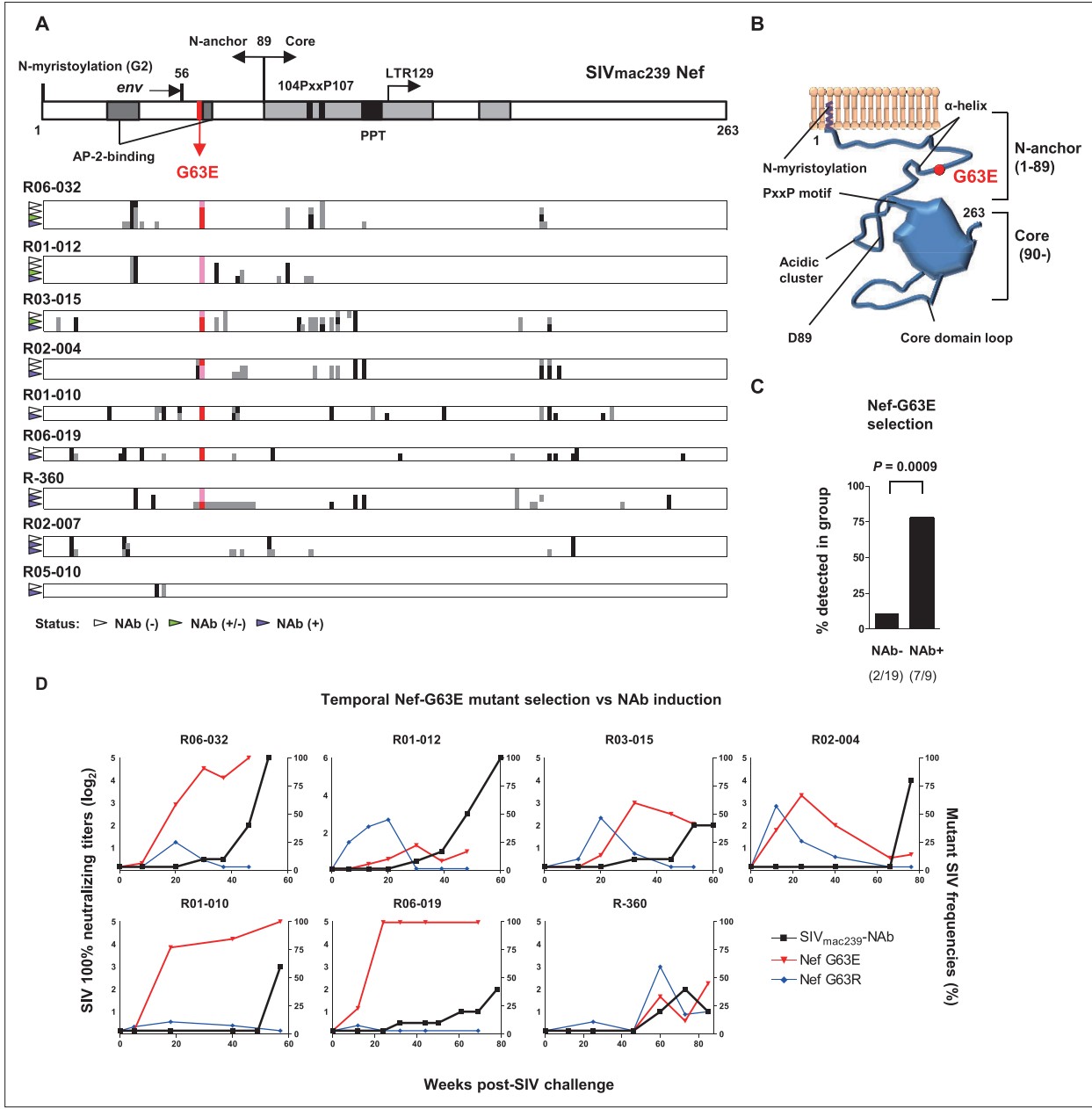

**Figure 2.** Selection of a viral *nef* mutation (Nef-G63E) before neutralizing antibody (NAb) induction. (**A**) Viral *nef* mutations in NAb inducers. A linear schema indicating Nef functional domains is aligned above. Mutations at time points with NAb⁻ (indicated by white wedge; mostly before 6 months post-infection [p.i.]), NAb⁺/⁻ (green wedge), and NAb⁺ (purple wedge; mostly after 1 year) are shown in individual animals. Black and dark gray represent dominant and subdominant mutations (or deletion in R-360) by direct sequencing, respectively. Red and pink indicate dominant and subdominant G63E detection, respectively. (**B**) Schema of SIV_{mac239} Nef structure and Nef-G63E mutation orientation. (**C**) Comparison of frequencies of macaques having Nef-G63E in plasma viruses between NAb non-inducers and inducers. Compared by Fisher's exact test. (**D**) Temporal relationship of Nef-G63E frequencies in plasma virus and NAb induction. Black boxes (left Y axis) show log_2 NAb titers; red triangles and blue diamonds (right Y axis) show percentage of G63E and G63R mutations detected by subcloning (15 clones/point on average), respectively.

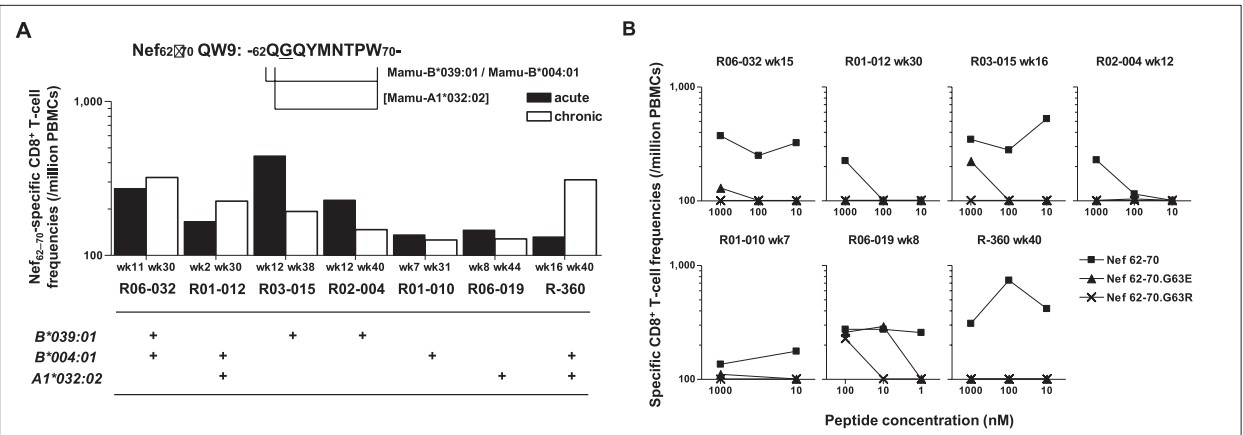

**Figure 3.** Nef-G63E is a CD8+ T-cell escape mutation. (**A**) Nef$_{62-70}$ QW9-specific CD8+ T-cell frequencies and related major histocompatibility complex class I (MHC-I) alleles in seven neutralizing antibody (NAb) inducers selecting Nef-G63E. Mamu-B*039:01 and Mamu-B*004:01 are known to restrict Nef$_{62-70}$ QW9 epitope and binding of Nef$_{63-70}$ peptide to Mamu-A1*032:02 was predicted. (**B**) CD8+ T-cell responses specific to wild-type (WT) Nef$_{62-70}$ or mutant Nef$_{62-70}$.G63E or Nef$_{62-70}$.G63R peptides.

The online version of this article includes the following figure supplement(s) for figure 3:

**Figure supplement 1.** Selection of Nef-G63E mutant simian immunodeficiency virus (SIV) is linked with Nef$_{62-70}$-specific CD8+ T-cell response positivity.

## Selection of a viral *nef* mutation (Nef-G63E) precedes chronic-phase SIV$_{mac239}$-specific NAb induction

To explore viral mutations linked to NAb induction, we assessed viral nonsynonymous polymorphisms outside *env*. Strikingly, we found selection of a viral genome mutation resulting in G (glycine)-to-E (glutamic acid) substitution at residue 63 of Nef (Nef-G63E) in seven of the nine NAb inducers (*Figure 2A*). Two inducers that did not select Nef-G63E were the early inducers detected for NAb positivity before 24 weeks p.i. This 63rd residue lies in the unstructured N-terminus of Nef, flanked by two α-helices conserved in SIVmac/SIVsmm (*Figure 2B*). This region is generally polymorphic among HIV-1/HIV-2/SIV, occasionally being deleted in laboratory and primary isolate HIV-1 strains (*Geyer et al., 2001*). AlphaFold2-based structure prediction did not derive palpable change except for low-probability disruption of the alpha helices (data not shown). This mutation was found only in 2 of the 19 NAb non-inducers, including one rapid progressor (*Figure 1—figure supplement 1*) and selection was significantly enriched in the NAb inducers compared to the 19 non-inducers (*Figure 2C*). Analysis of Nef-G63E mutation frequencies in plasma viruses showed that Nef-G63E selection preceded or at least paralleled NAb induction (*Figure 2D*).

## Nef-G63E is a CD8+ T-cell escape mutation

To explore the mechanistic link between Nef-G63E selection and NAb induction, first we assessed whether CD8+ T-cell responses target this region. We found that these seven NAb inducers selecting this mutation elicited CD8+ T-cell responses specific for a 9-mer peptide Nef$_{62-70}$ QW9 (QGQYMNTPW) (*Figure 3A*). This Nef$_{62-70}$ QW9 epitope is restricted by MHC-I molecules Mamu-B*004:01 and Mamu-B*039:01 (*Evans et al., 1999*; *Sette et al., 2012*). Possession of these accounted for six cases of Nef-G63E selection (*Figure 3A*), and the remaining one animal possessed Mamu-A1*032:02 predicted to bind to Nef$_{63-70}$ GW8 (NetMHCpan). When compared, 10 of 19 NAb non-inducers also possessed at least one of these alleles (*Figure 1—figure supplement 1*). This did not significantly differ with the NAb inducer group (p=0.25 by Fisher's exact test, data not shown), indicating that NAb induction was not simply linked with possession of these MHC-I genotypes but instead required furthermore specific selection of the Nef-G63E mutation (*Figure 2C*). Nef-G63E was confirmed to be an escape mutation from Nef$_{62-70}$-specific CD8+ T-cell responses (*Figure 3B*). NAb non-inducers possessing these alleles elicited little or no Nef$_{62-70}$-specific CD8+ T-cell responses (*Figure 3—figure supplement 1A*), suggesting that in vivo selection and fixation of this Nef-G63E SIV was indeed Nef$_{62-70}$-specific CD8+ T cell-dependent. Replication of SIV carrying the Nef-G63E mutation was comparable with WT on a cynomolgus macaque HSC-F CD4+ T-cell line (*Akari et al., 1996*), excluding the possibility that the

mutation critically impairs viral replication (*Figure 3—figure supplement 1B*), and plasma viral loads were comparable between Nef-G63E mutant-selecting NAb inducers versus non-inducers (*Figure 3—figure supplement 1C*). These results indicate NAb induction following in vivo selection and fixation of the CD8[+] T-cell escape *nef* mutation, Nef-G63E under viral persistence.

## G63E mutation reduces aberrant Nef interaction-mediated drive of PI3K/mTORC2

We next focused on the functional phenotype of Nef-G63E mutant SIV in infected cells. An essence of host perturbation by Nef is its wide-spectrum molecular downregulation, ultimately facilitating viral replication (*Schindler et al., 2004*; *Zhang et al., 2009*). To evaluate possible amelioration of this property, we compared downregulation of major targets CD3, CD4, MHC-I, CXCR4, and BST-2 (*Jia et al., 2009*) in infected HSC-F cells. Nef-G63E mutation did not confer notable change compared with WT (p=not significant [*n.s.*] for all molecules, data not shown), implicating other non-canonical changes (*Figure 4A*). One report suggested Nef to drive macrophage production of soluble ferritin and perturb B cells, with its plasma level correlating with viremia (*Swingler et al., 2008*). Here, plasma ferritin levels showed no differences (*Figure 4—figure supplement 1A*) and viral loads were comparable as aforementioned between Nef-G63E-selecting NAb inducers and non-inducers, arguing against gross involvement of ferritin as well as other viral replication-related Nef phenotypes at least in this model.

Next, we elucidated immunosignaling-modulating properties of Nef-G63E SIV. Akt is known as the predominant immune-intrinsic negative brake of B-cell maturation and AFC/antibody responses in vivo (*Omori et al., 2006*; *Srinivasan et al., 2009*; *Limon et al., 2014*; *Fruman et al., 1999*; *Ray et al., 2015*; *Sander et al., 2015*; *Luo et al., 2019*). Thus, we analyzed Akt phosphorylation on day 3 after Nef-G63E SIV infection at low multiplicity of infection (MOI). This analysis revealed that its serine 473-phosphorylated form (pAkt Ser473), non-canonically known for Nef-mediated upregulation (*Kumar et al., 2016*), was significantly lower in Nef-G63E mutant-infected cells compared with WT (*Figure 4B*). The difference observed here was more pronounced than histogram deviation levels in PI3K gain-of-function mice with full recapitulation of B-cell/antibody-dysregulating human APDS disease phenotype (*Avery et al., 2018*). In contrast, the threonine 308-phosphorylated Akt (pAkt Thr308) level remained unaffected (*Figure 4—figure supplement 1B*).

Interestingly, external PI3K stimuli by serum starvation (*Kennedy et al., 1997*) accelerated phenotype appearance from day 3 to day 1 p.i. (*Figure 4C*, left). A similar trend was obtained by short-term PI3K stimulation with IFN-γ (*Nguyen et al., 2001*), IL-2 (*Marzec et al., 2008*), and SIV Env (*François and Klotman, 2003*; *Figure 4—figure supplement 1C*). A high-MOI SIV infection, comprising higher initial concentration of extracellular Env stimuli, also accelerated phenotype appearance from day 3 to day 1 p.i., with stronger pAkt reduction (*Figure 4C*, right). These data indicate that the Akt-inhibitory G63E mutant Nef phenotype is PI3K stimuli-dependent. Transcriptome analysis signatured decrease in PI3K-pAkt-FoxO1-related genes in mutant SIV infection (*Figure 4—figure supplement 1D*). Given that Akt is also a survival mediator, Nef-G63E mutation may be T helper cell (Th)-cytopathic and decrease Th dysfunctional expansion (*Baumjohann et al., 2013*), as had been observed in LCMV strain WE (*Recher et al., 2004*) and chronic HIV/SIV (*Gray et al., 2011*; *Lindqvist et al., 2012*; *Petrovas et al., 2012*) infections. Here, reduced chronic-phase peripheral CXCR3⁻CXCR5⁺PD-1⁺ memory follicular Th (Tfh), linked with antibody cross-reactivity in one cohort (*Locci et al., 2013*), was somewhat in line with this notion (*Figure 4—figure supplement 1E and F*). We additionally examined another CTL escape mutant Nef-G63R, selected with marginal statistical significance primarily in early stage (*Figure 4—figure supplement 2A*). This mutant did not associate with NAb induction in one control animal (R06-034) even upon preferential selection (*Figure 4—figure supplement 2B*) and Nef-G63R mutant was not similarly decreased in pAkt drive (*Figure 4—figure supplement 2C*), implicating that the Nef-G63E mutation was more tightly linked with NAb induction in terms of an immunosignaling phenotype.

We further explored molecular traits of this decreased Akt hyperactivation. We reasoned that comparing endogenous Nef binding patterns would be adequate and analyzed SIV-infected HSC-F cells with a flow cytometry-based proximity ligation assay (PLA) (*Leuchowius et al., 2009*; *Avin et al., 2017*). We found that this Nef-G63E mutation causes significant decrease in Nef binding to PI3K p85/p110α and downstream mTORC2 components mTOR (*Sarbassov et al., 2005*) and GβL/mLST8 (*Kim et al., 2003*) in the CD4[lo]-SIV Gag p27[+] infected population (*Figure 4D*). Results were biochemically

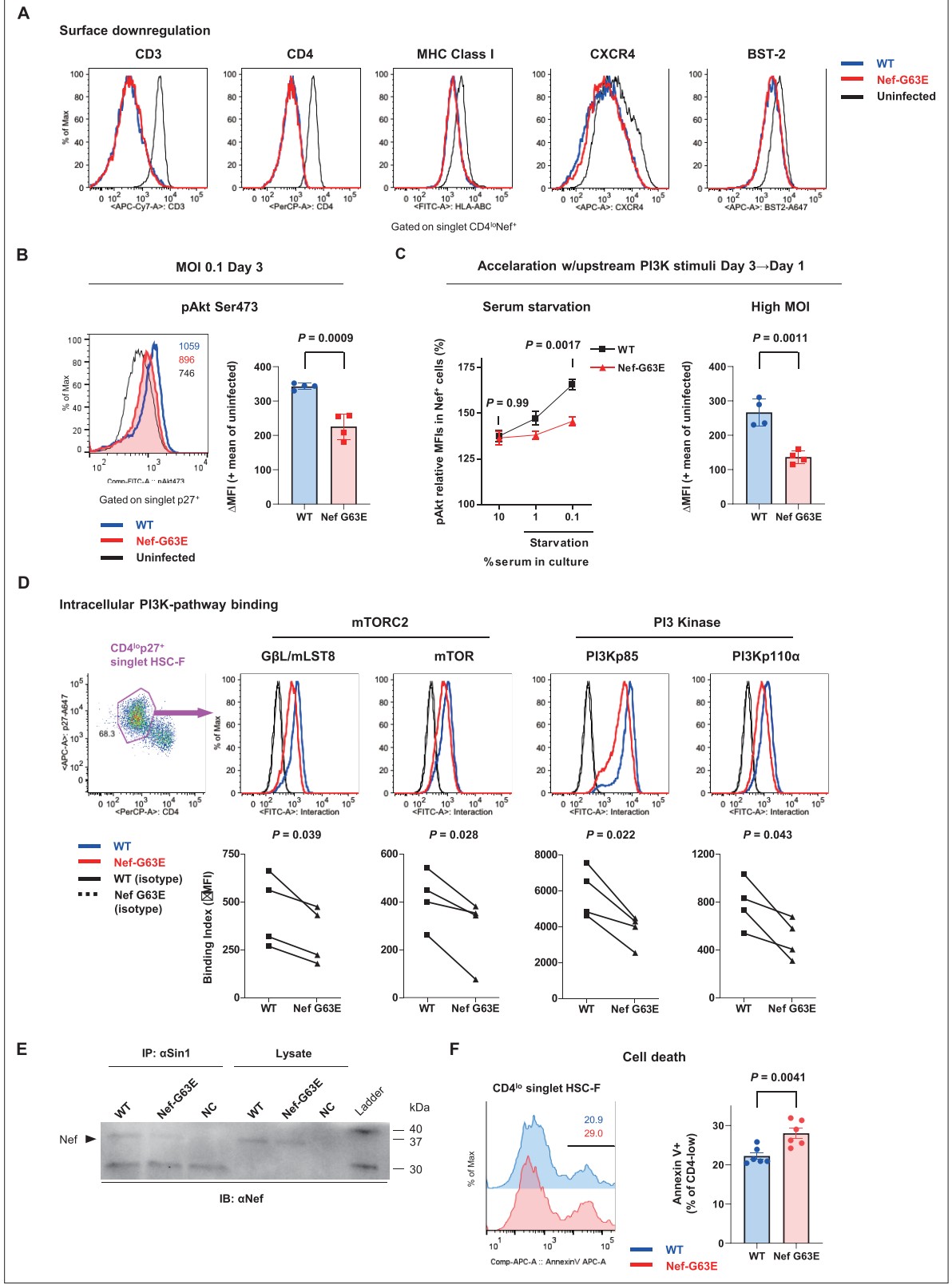

**Figure 4.** Nef-G63E mutation reduces PI3K/mTORC2 binding and pAkt drive. (**A**) Representative surface expression level histograms of CD3, CD4, major histocompatibility complex class I (MHC-I), CXCR4, and BST-2 in CD4$^{lo}$Nef$^+$ subpopulations after wild-type (WT) or Nef-G63E mutant simian immunodeficiency virus (SIV) infection at multiplicity of infection (MOI) 0.1 on HSC-F cells. (**B**) Left: representative histograms of relative pAkt serine (Ser) 473 levels in p27$^+$ subpopulations after WT or Nef-G63E mutant SIV infection at MOI 0.1 on HSC-F cells. Numbers show pAkt Ser473 mean fluorescence

*Figure 4 continued on next page*

Figure 4 continued

intensities (MFIs) for each. Right: deviation of pAkt Ser473 MFIs in p27$^+$ HSC-F cells compared with mean MFI of uninfected cells. Compared by unpaired $t$-test. (C) Left: relative pAkt Ser473 levels (normalized to mean MFI of uninfected controls) in Nef$^+$ HSC-F cells assessed for serum starvation (MOI 0.2, 1 day post-infection [p.i.]). Adjusted p values show results of comparison via Sidak's post hoc test of two-way ANOVA (C, left). Right: deviation of pAkt Ser473 MFIs in Nef$^+$ HSC-F cells assessed for high-MOI infection (MOI 5, 1 day p.i). Compared by unpaired $t$-test. (D) Proximity ligation assay (PLA) of Nef binding with mTOR, GβL/mLST8, PI3Kp85, and PI3Kp110α. MFI-based binding index was calculated as (anti-Nef/anti-partner) - (isotype/anti-partner) – (anti-Nef/isotype) + (isotype/isotype). Histograms for samples and isotype/isotype are representatively shown. Differences in MFI binding indexes can be enhanced compared with comparison of raw MFIs. Compared by paired $t$-tests. (E) Sin1 co-immunoprecipitation analysis of WT versus Nef-G63E mutant SIV-infected HSC-F cells (infected at MOI 0.05, day 3). Immunoblotting of Nef (37 kDa) in whole-cell lysates (lanes 4–6) and anti-Sin1 antibody immunoprecipitates (lanes 1–3) are shown. (F) Cell death frequencies of SIV-infected cells measured by Annexin V positivity (% of CD4$^{lo}$) (infected at MOI 0.1, day 3). Compared by unpaired $t$-test. Data represent one of two (A–C, right) independent experiments in quadruplicate, four independent experiments performed in triplicate (C, left), four independent single-well comparison experiments pooled for statistical analysis (D), two experiments (E) or two experiments performed with six wells/control (F). Bars: mean ± SD (B, C right), mean ± SEM (C left, F).

The online version of this article includes the following source data and figure supplement(s) for figure 4:

**Source data 1.** Original western blots for *Figure 4E* indicating the relevant bands and treatments.

**Source data 2.** Original file for the western blot analysis displayed in *Figure 4E*.

**Figure supplement 1.** Decreased Akt hyperactivation properties of Nef-G63E mutant simian immunodeficiency virus (SIV).

**Figure supplement 2.** Nef-G63R mutant simian immunodeficiency virus (SIV) properties.

---

confirmed by a decrease in G63E-mutated Nef binding to the mTORC2-intrinsic cofactor Sin1 in coimmunoprecipitation analysis of infected HSC-F cells (*Figure 4E*). Collectively, the Nef-G63E mutation attenuates PI3K/mTORC2 signaling driven by aberrant Nef bridging, explaining decreased Akt Ser473 phosphorylation by mTORC2. These properties were in keeping with more pronounced apoptosis of CD4-downregulated/infected HSC-F cells upon Nef-G63E SIV infection (*Figure 4F*). Thus, Nef-G63E SIV is a mutant virus decreased in aberrant interaction/drive of B-cell-inhibitory PI3K/mTORC2 signaling, manifesting a molecular signature reciprocal to human APDS (*Angulo et al., 2013*; *Lucas et al., 2014*; *Avery et al., 2018*).

## Targeting of lymph node Env-specific B cells by Nef in vivo

Next, we analyzed in vivo targeting of virus-specific B cells by Nef in lymph nodes to explore the potential B-cell-intrinsic influence of the Nef-G63E phenotype. Previously suggested influence of soluble Nef itself (*Qiao et al., 2006*) and/or related host factors (*Swingler et al., 2008*) may derive generalized negative influence on B cells. However, SIV antigen-specific binding antibody responses were rather decreased in NAb inducers (*Figure 1—figure supplement 1*), with comparable viral loads (*Figure 3—figure supplement 1C*) and ferritin levels (*Figure 4—figure supplement 1A*), differing from the aforementioned literature stating positive correlation between the three. We surmised that some targeted Nef intrusion against Env-specific B cells may be occurring and that the decreased aberrant drive of Nef-PI3K/mTORC2 may result in their enhanced maturation in lymph nodes.

While reports (*Xu et al., 2009*) histologically proposed Nef B-cell transfer, quantitative traits, for example, invasion frequency and influence on virus-specific B cells, have remained unvisited. One of the reasons is that intracellular Nef staining is dim (particularly for B cell-acquired Nef) and difficult to examine by conventional flow cytometry. To circumvent this issue, defining staining cutline using single-cell images potentially overcomes confounding technical hurdles, such as high Nef false-staining signals owing to pre-permeation rupture and post-permeation processing that derives biologically discontinuous staining and/or batch-inflated signals. Thus, we reasoned that at the expense of spatial information, sophisticating imaging cytometry would best visualize Nef-mediated B-cell perturbation in vivo. We analyzed lymph node B cells with Image Stream X MKII, with high-power (>10-fold) antigen detection (e.g., molecules of equivalent soluble fluorochrome [MESF] 5 vs. MESF 80 in an average flow cytometer for FITC detection) ideal for detection/single-cell verification.

We designed a triple noise cancellation strategy to overcome the issues stated above. Firstly, amine reactive dye staining (*Perfetto et al., 2006*) gated out B cells being Nef-positive due to pre-experimental membrane damage (*Figure 5—figure supplement 1A*, left). Next, we deployed secondary quantitative parameters of Nef signals deriving from each pixel of the images. Generation of a gray-level cooccurrence matrix (GLCM) (*Haralick et al., 1973*), which computes adjacent signal deviation as their frequencies on a transpose matrix, enables calculation of a variety of feature values

summarizing traits of the whole image. A biological assumption of Nef signal continuity in a true staining suggested that the sum of weighted square variance of GLCM, that is, the {sum of [square of (value – average signal strength)×frequency of each value occurrence]} would separate natural versus artificial Nef signals. This value, known as Haralick variance, was computed for multiple directions and averaged. The resultant Haralick variance mean (*Figure 5—figure supplement 1A*, X-axis) is proportionate with unnaturalness of signals in the Nef channel. Finally, Nef signal intensity threshold (*Figure 5—figure supplement 1A*, Y-axis) gated out overtly stained cells void for true-false verification. These measures excluded B cells with non-specific, strong-signal binary-clustered staining pixels for Nef (deriving a large summated variance: X-axis and/or batch-stained for Nef: Y-axis; likely originating from post-experimental membrane damage).

Using this approach, we successfully acquired images of viable Nef$^+$CD19$^+$ B cells with low Nef signal Haralick variance mean and low Nef signal intensity threshold, with fine-textured gradation of low-to-intermediate Nef staining with continuity from membrane-proximal regions and without sporadic staining speckles (*Figure 5*, *Figure 5—figure supplement 1*). Scoring segregation of typical void/valid images by a linear discriminant analysis-based machine learning module showed that this gating provides the highest two-dimensional separation (*Figure 5—figure supplement 1*). Combined with visualization of Env-binding B cells, we reproducibly obtained images of Env-specific (Ch 12) CD19$^+$ (Ch 11) B cells without membrane ruptures (Ch 08) and showing fine-textured transferred Nef (Ch 02) (*Figure 5B*). Nef invasion upregulated pAkt Ser473 (Ch 03) to a range resembling in vitro analysis (*Figure 5C*), demonstrating that Nef-driven aberrant PI3K/mTORC2 signaling does occur in Nef-invaded B cells in vivo. Strikingly, lymph node Env-specific B cells showed significantly higher Nef-positive frequencies as compared with batch non-Env-specific B cells (*Figure 5D*). This indicated that infected cell-derived Nef preferentially targets adjacent Env-specific B cells, putatively through its contact/transfer from infected CD4$^+$ cells to B cells (*Xu et al., 2009*; *Hashimoto et al., 2016*). Thus, the phenotypic change in Nef can directly dictate Env-specific B-cell maturation.

## Cell contact-dependent B-cell invasion from infected cells in vitro

To understand cell-intrinsic properties enhancing B-cell Nef acquisition in vivo, we addressed in an in vitro coculture reconstitution how infected cell-to-B-cell Nef invasion takes place and how it is modulated. We performed imaging cytometry on a 12-hour coculture of SIV-disseminating HSC-F cells (reaching around 30% Nef-positivity at moment of coculture) with Ramos B cells, with or without modulators added throughout the coculture period (*Figure 6A*, top left). In this coculture, there is no MHC-related interaction known to enhance T-cell/B-cell contact (*Wülfing et al., 1998*). Machine learning-verified noise cancellation of fragmented signal acquisition (*Figure 6A*, middle left) filtered reproducible images of Nef-positive HSC-F cells directly adhering to Ramos B cells within the doublet population (*Figure 6A*, top to middle right). Importantly, addition of soluble, stimulation-competent antibodies against macaque CD3 enhanced adhesion of Nef$^+$ T cells to B cells (*Figure 6A*, bottom center). In these doublets, we readily detected images of polarized Nef protrusion/injection from T cells to B cells (second column, filled wedges) as well as trogocytosed B-cell membranes (third column, open wedge), suggesting dynamic Nef acquisition by B cells. Changing centrifugation strengths upon coculture collection and staining (190 × *g* vs. 1200 × *g* and 600 × *g* vs. 1200 × *g*, respectively) shifted the tendency between enriched Nef$^+$ HSC-F-B-cell doublet versus singlet detection via CD3 stimulation, respectively (*Figure 6A*, lower right).

Substantiated by the acquired images, an enhancement of live singlet Ramos B-cell Nef acquisition (% Nef$^+$ in total B cells) was also observed by CD3 stimulation in conventional flow cytometry (*Figure 6B*). There was a similar enhancement when we added phytohemagglutinin (PHA), whereas phorbol-12-myristate-13-acetate (PMA)/ionomycin did not affect Nef acquisition by B cells. In a similar 72-hour static coculture with primary CD20$^+$ B cells, we detected recapitulation of the decreased B-cell pAkt Ser473 phosphorylation upon G63E Nef transfer compared to that of WT Nef (*Figure 6C*), indicating that the Nef-G63E mutation can directly alleviate maturation inhibition in preferentially Nef-targeted Env-specific B cells.

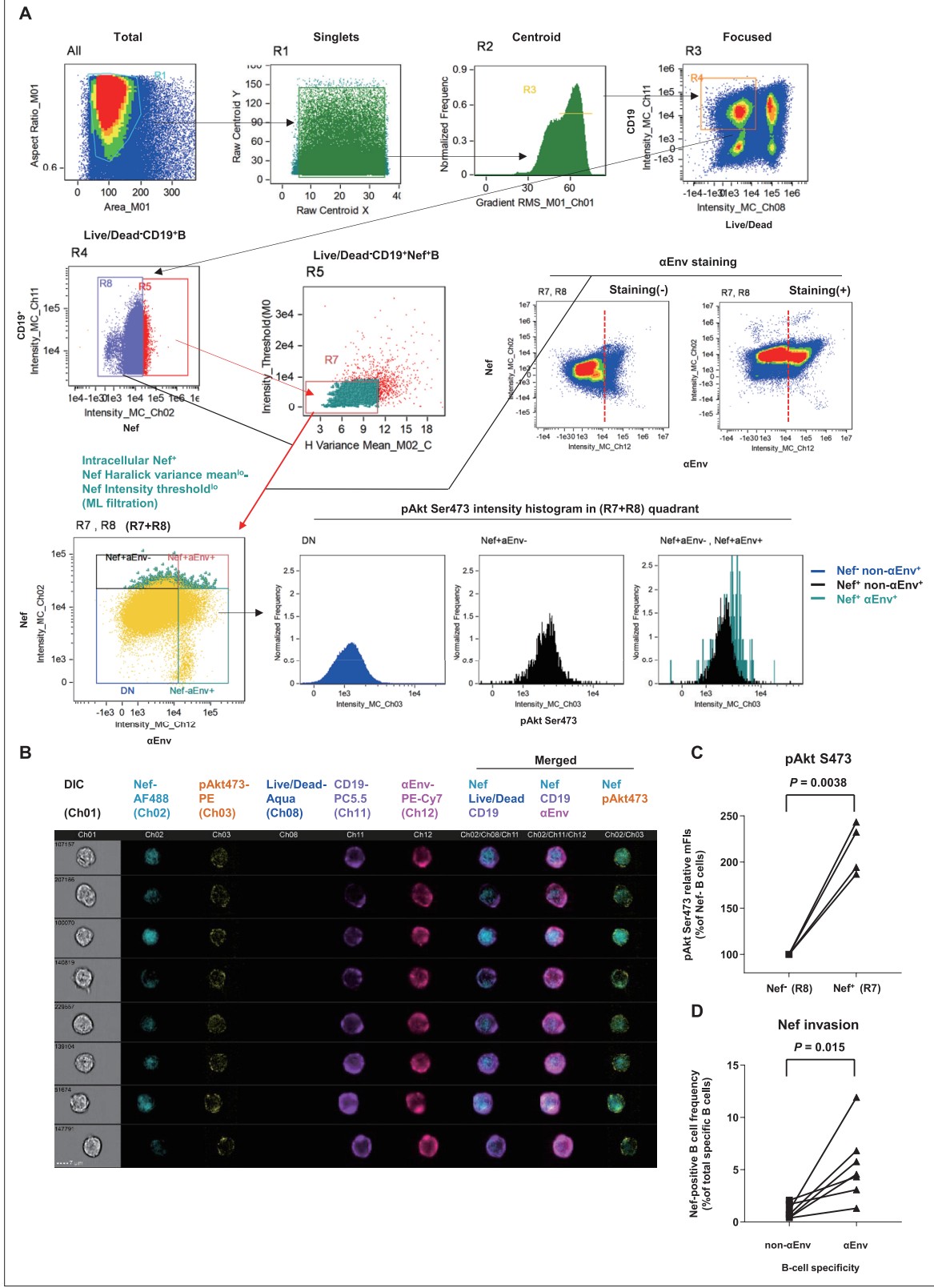

**Figure 5.** Preferential targeting of lymph node Env-specific B cells by Nef in vivo. (**A**) Representative gating of triple noise cancellation in vivo Nef staining in B cells analyzed using ImageStreamX MKII. Pre-experimental damaged cells are first excluded with Live/Dead from focused/centroid/singlet image-acquired CD19+ B cells (first lane right/R4 gated on 'Focused'). Following Nef+ gating (second lane left/R5 gated on R4), a second step of Nef noise cancellation (second lane middle/R7 on R5) comprises double-negative removal of post-experimental stochastic irregular staining building a

*Figure 5 continued on next page*

Figure 5 continued

disparate intracellular staining gradient (X-axis, Nef signal pixel Haralick variance mean) and post-experimental batch overt cellular staining (Y-axis, Nef signal pixel intensity threshold). This outputs a B-cell population with a fine-textured pericellular Nef$^{int-lo}$ staining, biologically concordant with Nef membrane-anchoring. Probing of anti-Env BCR (αEnv) by recombinant SIV Env (second lane, right) is combined, resulting in a 2-D panel of intracellular Nef versus αEnv for noise-cancelled Nef$^+$ B cells (R7) plus Nef$^-$ B cells (R8) (third lane left/'R7+R8'). pAkt Ser473 expression (third lane right) and cellular morphology (B) was further analyzed. DN, double-negative. (B) Typical images of Nef-transferred Env-specific B cells defined αEnv$^+$-intracellular Nef$^+$-Nef Haralick variance mean$^{lo}$-Nef Intensity threshold$^{lo}$-Live/Dead$^-$-CD19$^+$ cells ('Nef$^+$aEnv$^+$' population of the lower-left panel in A, gated on 'R7, R8'). Note the pericellular pAkt Ser473 upregulation in these cells (Ch 03/yellow). Data on inguinal lymph node lymphocytes of macaque R10-007 at week 62 post-SIV$_{mac239}$ infection are shown in (A) and (B). (C) Comparison of pAkt Ser473 median fluorescence (medFI) intensity levels in Nef$^-$ B cells (R8) versus noise-cancelled Nef$^+$ B cells (R7). Analyzed by paired $t$-test. (D) Comparison of Nef-positive cell frequencies in non-Env-specific (left) versus Env-specific (right) B cells in lymph nodes of persistently SIV-infected macaques (n = 6). Analyzed by paired $t$-test.

The online version of this article includes the following figure supplement(s) for figure 5:

Figure supplement 1. Machine learning-verified morphological gating of Nef-invaded B cells in vivo.

## Enhanced Env-specific B-cell responses after PI3K-diminuting mutant selection

Finally, to assess in vivo B-cell quality in NAb inducers with Nef-G63E mutant selection, we examined peripheral SIV Env-specific IgG$^+$ B-cell responses comprising plasmablasts (PBs) and memory B cells (B$_{mem}$). After excluding lineage-specific cells (T/NK/pro-B cells/monocytes/myeloid dendritic cells [DCs]/plasmacytoid DCs), we defined IgG$^+$ PBs as showing high replication (Ki-67$^+$), post-activation (HLA-DR$^+$), transcriptional switching for terminal differentiation (IRF4$^{hi}$), and downregulated antigen surface binding (surface [s]Env$^{lo}$-cytoplasmic [Cy]Env$^+$) (*Nutt et al., 2015*; *Silveira et al., 2015*; *Figure 7*, *Figure 7—figure supplement 1*). NAb inducers showed significantly higher Env-specific IgG$^+$ PB responses around week 30 p.i., after Nef-G63E selection, compared to those in NAb non-inducers (*Figure 7B and C*). At 1 year, this difference became enhanced as a result of less pronounced decrease in Env-specific PB frequencies in the NAb inducers.

Simultaneous quantification of Env-specific PB and IgD$^-$CD27$^+$IgG$^+$ B$_{mem}$ responses allowed for an assessment of overall B-cell response quality by pair-wise analysis of Env-specific IgG$^+$ B$_{mem}$/PB as a projection of germinal center (GC) output (*Zotos et al., 2010*; *Figure 7D*, left). In this two-dimensional vector temporally plotting the frequency of B$_{mem}$ and PBs, vector protrusion toward the upper right represents higher gross GC output of antibody-forming cells (AFCs). In NAb non-inducers, all vectors converged on an empirically defined polygonal attractor area $D_n$ (gray area surrounded with dotted lines in *Figure 7D*, right) at year 1 p.i. and beyond, describing that these NAb non-inducers failed to sustain Env-specific GC output. In contrast, the vectors were consistently tracked outside $D_n$ in the NAb inducers with Nef-G63E (*Figure 7D*). At the time of NAb induction, they converged on another upper-right GC output attractor area $D_i$ (red area surrounded with dotted lines in *Figure 7D*, right) that is mutually exclusive with $D_n$ (p<0.0001 by Fisher's exact test on NAb non-inducer/inducer vector distribution frequency within $D_n$). These suggest more robust virus-specific IgG$^+$ B-cell responses following Nef-G63E CD8$^+$ T-cell escape mutant selection, ultimately leading to NAb induction. The enhanced signature of cognate B cells was also an inverted pattern of impaired terminal differentiation of B-cell responses in APDS (*Al Qureshah et al., 2021*).

Taken together, in the current model, Nef$_{62-70}$-specific CD8$^+$ T-cell responses in persistently SIV$_{mac239}$-infected macaques selected for an escape mutant, Nef-G63E, which displays attenuated aberrant Nef binding and ensuing drive of B-cell-inhibitory PI3K/mTORC2. Nef invasion of B cells in vivo occurred more preferentially in Env-specific B cells, suggesting diminution in Nef-mediated tonic dysregulation of B cells after mutant selection. These events conceivably predisposed to enhanced Env-specific B-cell responses and subsequent SIV$_{mac239}$-specific NAb induction, altogether, in a manner reciprocal to human APDS-mediated immune dysregulation.

## Discussion

In the present study, we found that in macaques infected with an NAb-resistant SIV, selection of a CD8$^+$ T-cell escape *nef* mutant virus, Nef-G63E, precedes NAb induction. As a result, Nef binding-mediated drive of PI3K/mTORC2 in Env-specific B cells becomes attenuated, which in turn unleashes maturation of antiviral B-cell responses in vivo to induce NAbs. Importantly, this manifested through

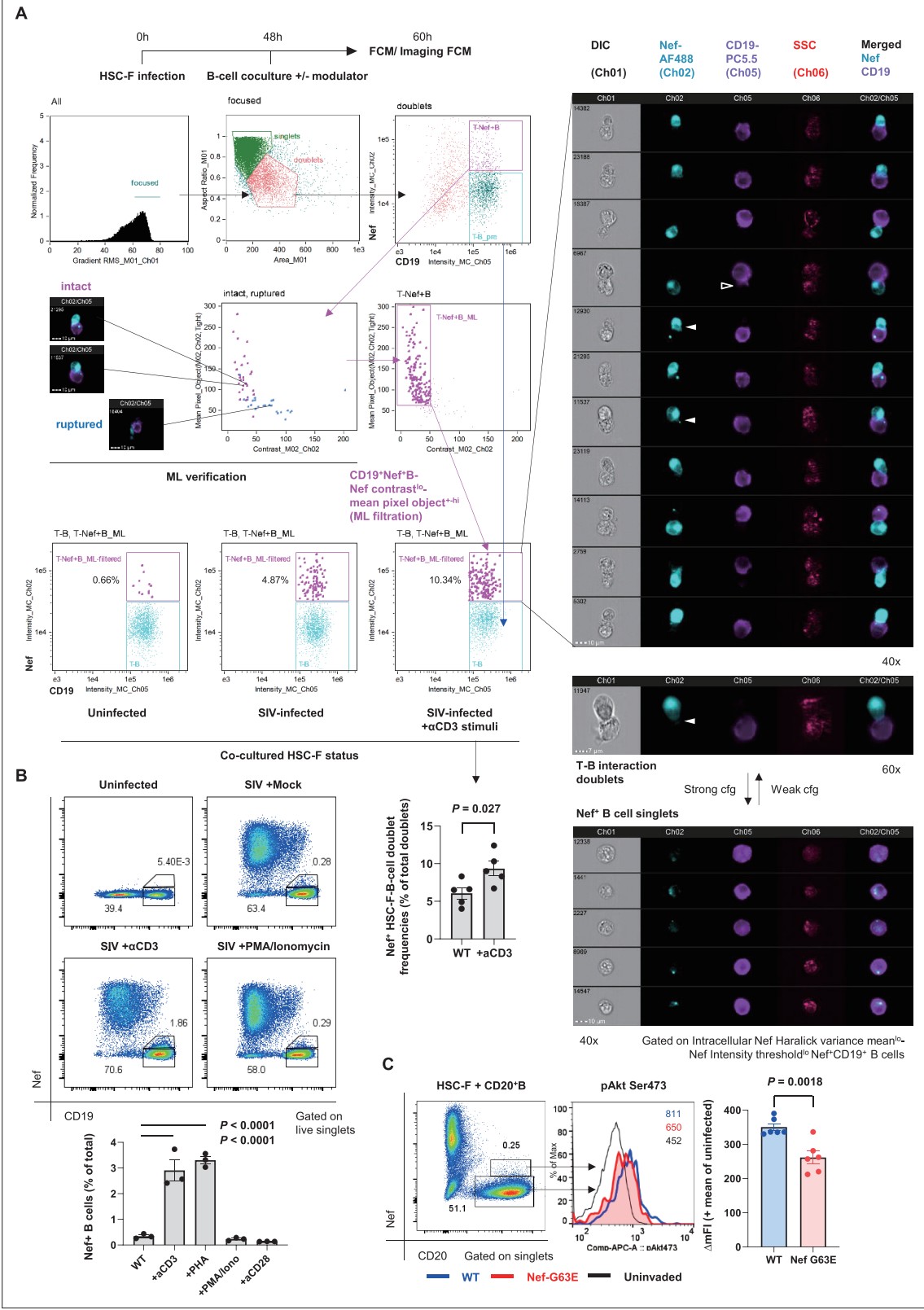

**Figure 6.** Contact-dependent enhancement of B-cell Nef acquisition in reconstitution. (**A**) Imaging flow cytometric analyses of stimuli-dependent enhancement of infected HSC-F cell adhesion to non-permissive CD19⁺ Ramos B cells with the indicated gating (left). Cocultured doublet cells subjected to machine learning-verified filtration of fragmented signal acquisition (via gating Nef pixel signal contrast^lo^-Nef mean pixel per object^hi^ cells) deriving typical images of infected HSC-F cell (Nef stain shown in pale blue pseudocolor) adhesion to Ramos cells (CD19 stain shown in purple

*Figure 6 continued on next page*

*Figure 6 continued*

pseudocolor) are shown. Analyzed by unpaired *t*-test for doublet formation in the unstimulated versus anti-CD3 monoclonal antibody (clone FN-18)-stimulated group. (**B**) Flow cytometric analyses of Nef-acquiring live CD19⁺ Ramos B cells upon coculture with simian immunodeficiency virus (SIV)-infected HSC-F cells with or without the indicated stimuli. Analyzed by one-way ANOVA with Tukey's post hoc multiple comparison tests. (**C**) Representative flow cytometric plot (left), histogram (middle), and pAkt Ser473 signal deviation in WT versus G63E Nef-acquiring CD20⁺ primary B cells (six wells/control). Analyzed by unpaired *t*-test. Data represent pooled data of two experiments (**A**) or one of three (**B, C**) independent experiments with indicated number of replicates showing similar results.

a pAkt deviation level more pronounced than WT versus germline *PIK3CD* gain-of-function mutation heterozygote mice recapitulating full human APDS phenotype (*Avery et al., 2018*). Thus, the current Nef-G63E-associated B-cell/NAb phenotype likely occurs in a 'reciprocal APDS-like' manner. This proposition was enhanced based on the extension of immune cell-intrinsic Nef influence on cognate B cells (*Figure 5*), in addition to infected T cells. This work is, to our knowledge, the first to interlink a PI3K/mTORC2-modulating viral signature and enhanced B-cell/NAb responses in a viral infection model.

A link between viral T-cell escape and consequent immune modulation has been previously explored to some extent. For example, enhanced binding of mutant HIV-1 epitope peptide to inhibitory MHC-I on DCs impairs T cells (*Lichterfeld et al., 2007*) and decreased CTL-mediated lymph node immuno-pathology can drive, and not inhibit, the production of LCMV-specific antibodies (*Battegay et al., 1993*). Our results now evidence a new pattern of NAb responses that are bivalently shaped through viral interactions with both humoral and cellular immunity in AIDS virus infection. SIV$_{mac239}$ infection of macaques possessing MHC-I alleles associated with Nef-G63E mutation can be one unique platform to analyze virus–host interaction for B cell maturation leading to NAb induction. The interval between Nef-G63E mutant selection and NAb induction was variable in the NAb inducers, for which we did not obtain a clear explanation. This may be influenced by certain basal competitive balance between humoral versus cellular adaptive immunity (*Recher et al., 2007*), as observed in certain vaccination settings (*Querec et al., 2009*).

The current Nef-G63E phenotype identified here adds yet another aspect to the wealth of evidence documenting the multifaceted impact of Nef in HIV/SIV infection (*Kirchhoff et al., 1995*; *Gauduin et al., 2006*; *Stolp et al., 2012*). Based on our data, the current phenotype differed from decreased replication-related properties or generalized amelioration in immune impairment such as those of delta-Nef HIV/SIV (*Kirchhoff et al., 1995*; *Johnson et al., 1997*; *Gauduin et al., 2006*; *Fukazawa et al., 2012*). The SIV Nef N-terminal unstructured region comprising Nef-G63E is not conserved in HIV-1 (*Schindler et al., 2004*), and beyond this model study it remains to be addressed whether a mutant HIV-1 with a similar immunosignaling-related phenotype can be obtained and how much lenti-viral conservation exists for such interactions. Analysis of cognate B-cell maturation on cohort basis (*Hau et al., 2022*) may potentially assist this approach.

Preferential Nef transfer to Env-specific B cells (*Figure 5*) suggests potential involvement of polar-ized perturbation in cognate immune cells in vivo, which was best depicted for preferential infectivity of HIV-specific CD4⁺ T cells (*Douek et al., 2002*) and HIV condensation to the interface of DC-T-cell contacts (*McDonald et al., 2003*). As cognate MHC class II interactions (*Wülfing et al., 1998*; *Gitlin et al., 2014*) regularly occur between HIV/SIV-specific CD4⁺ T cells and Env-specific B cells in GCs, dysregulation of Env-specific B cells may well result from hijacking of cognate T-cell/B-cell interactions by the virus. Enhanced T-cell attachment and transfer of Nef to B cells by TCR stimulation (*Figure 6*) may override constraints of infection-mediated CD4⁺ T-cell cytoskeletal impairment (*Campbell et al., 2004*; *Jolly et al., 2004*; *Stolp et al., 2009*), particularly by antagonizing reduced motility (*Stolp et al., 2012*), resulting in in vivo 'filtering' of enhanced cognate CD4⁺ T-cell interaction with cognate B cells. Interestingly, Tfh cells that bear trogocytosed CD20 and are positive for viral DNA appear in vivo during SIV infection and increase when infection progresses (*Samer et al., 2023*). This is consistent with our data that suggest occurrence of contact-dependent, polarized interactions facilitating Nef transfer to cognate B cells.

The major in vivo readout of this study is autologous neutralization of a highly difficult-to-neutralize SIV strain, which is different from HIV-specific broadly neutralizing antibodies (bNAbs). Importantly, however, obtained signatures of enhanced IgG⁺ Env-specific AFC and B cells (class switch/terminal differentiation) and virus neutralization (hypermutation) in our model are indicative of elevated activity

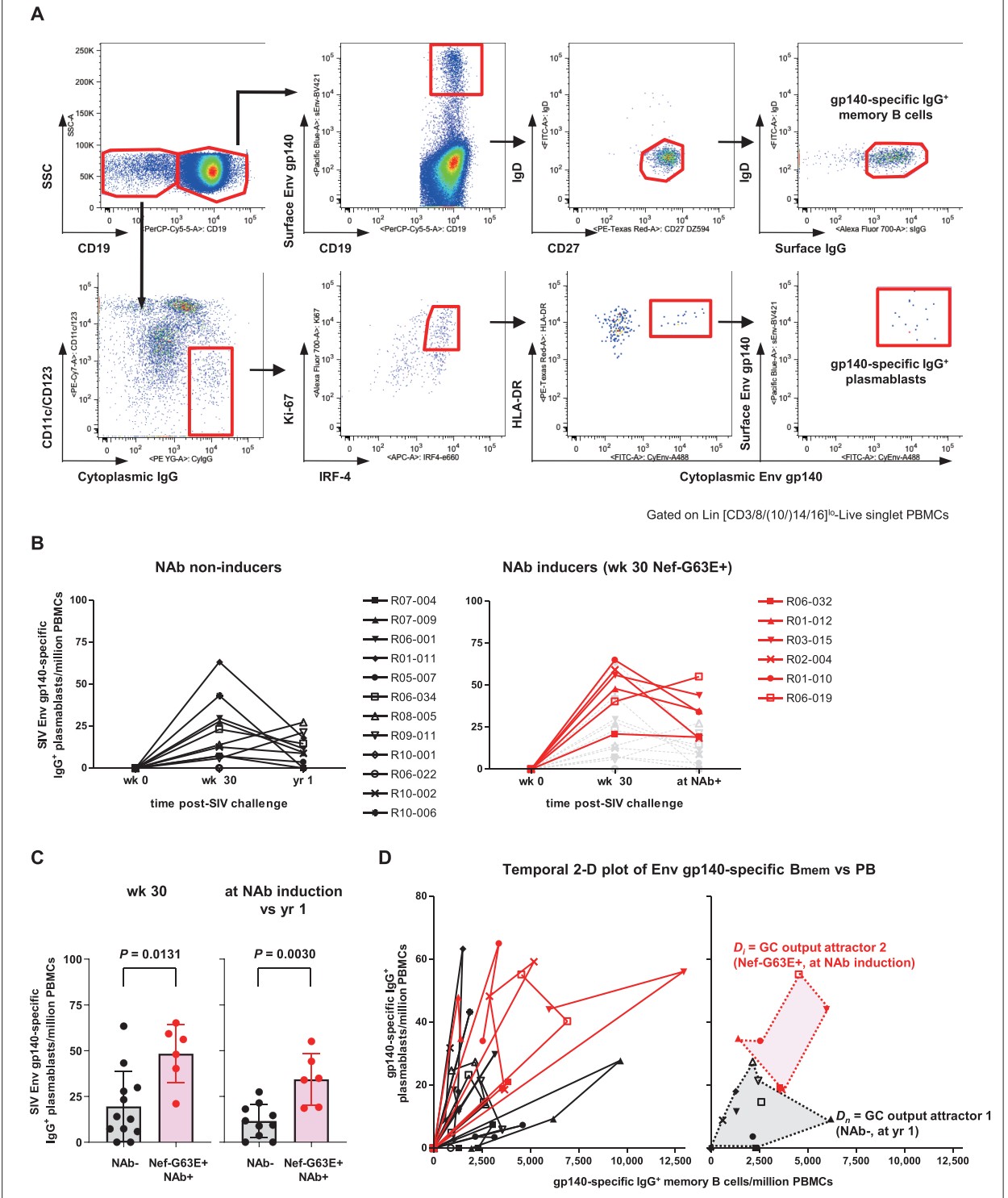

**Figure 7.** Enhanced simian immunodeficiency virus (SIV) Env-specific B-cell output up to neutralizing antibody (NAb) induction following Nef-G63E selection. (**A**) Representative gating (R02-004, week 32 post-infection [p.i.]) of SIV Env gp140-specific memory B cells (Bmem) and plasmablasts (PBs). Two panels for PB gating are shown in lower resolution for visibility. Bmem staining was performed separately and gating are merged with PB panels (first row, panels 2–4). (**B**) Changes in SIV Env gp140-specific PB frequencies in viremic NAb non-inducers (left, n = 12) and Nef-G63E-selecting NAb inducers (right, n = 6). Available samples of twelve viremic NAb non-inducers (including ten with ≥1-year survival) and six NAb inducers were tracked. In the right panel, the frequencies in NAb non-inducers are shown in background (gray) for comparison. (**C**) Comparison of Env gp140-specific PB frequencies between NAb non-inducers and inducers by Mann–Whitney *U* tests. In the right, the frequencies at NAb induction were compared with those at year 1 in NAb non-inducers with ≥1-year survival (n = 10). Bars: mean ± SD. (**D**) Left: vector chart of Env gp140-specific memory B cell (Bmem) and PB levels.

*Figure 7 continued on next page*

*Figure 7 continued*

Legends for each animal correspond to the ones in (**B**). Right: NAb non-inducer vectors empirically define a polygonal GC output attractor area $D_n$ (gray area surrounded with dotted lines) on which they converge by and beyond year 1 p.i. The NAb inducer vectors (shown up to the time of NAb induction) remained outside of $D_n$. At the moment of NAb induction they converged on a second GC output attractor area $D_i$ (red area surrounded with dotted lines), mutually exclusive with $D_n$ (p<0.0001 by Fisher's exact test on NAb inducer vector convergence frequency within $D_n$). Legends for year 1 p.i. in the NAb non-inducers and moment of NAb induction in the NAb inducers are specified.

The online version of this article includes the following figure supplement(s) for figure 7:

**Figure supplement 1.** Full gating strategy of SIV$_{mac239}$ Env gp140-specific B-cell responses.

of activation-induced cytidine deaminase (AID), the canonical positive driver of B-cell fate. Thus, we reason that findings in this study have a strong conceptual continuity with bNAb regulation by AID in HIV infection.

Our results extracted an in vivo link between a decrease in aberrant Nef-PI3K/mTORC2 interaction and major enhancement in B-cell responses. The relationship is reciprocal to immunogenetic mechanisms of human PI3K gain-of-function mutations, resulting in APDS with multiply impaired antiviral B-cell responses. APDS occurs from germline mutations in leukocyte-intrinsic p110δ catalytic subunit-coding *PIK3CD* or the ubiquitous p85α regulatory subunit-coding *PIK3R1*. In both cases, the mutations cause residue substitutions, resulting in class IA PI3K gain of function (*Dornan et al., 2017*). Hyper-activated PI3K interferes with cognate B cells in an immune cell-intrinsic manner, leading to impaired B-cell class switching and resultant susceptibility against various infections (*Angulo et al., 2013*; *Lucas et al., 2014*; *Coulter et al., 2017*). In the current work, we conversely document a decrease in virally induced tonic PI3K drive, enhanced IgG class-switched cognate B-cell responses and NAb induction against a notably difficult-to-neutralize SIV. The exact hierarchy of binding between Nef and PI3K/mTORC2 components (including PI3K isoforms) and its alteration by Nef-G63E mutation remains to be investigated. Yet PI3Kp85 showed the strongest Nef-binding index in PLA assay (*Figure 4D*) presumably because of the SH3-binding PxxP domain in Nef, which binds the SH3 domain of PI3Kp85 (*Rickles et al., 1994*). Decreased binding of Nef to its canonical target PI3Kp85 may precipitate into attenuated interactions of Nef with quintuple PI3K/mTORC2, resulting in fragile downstream Akt signaling. Such domino-triggering is also observed in APDS patients, from gain-of-function mutant p85 to p110 (*Dornan et al., 2017*), which similarly illustrates the fine-tuned nature of the PI3K/mTORC2/Akt signaling pathway.

A limitation of this study is the use of retrospective samples, posing constraints for detecting exact temporal changes in tonic viral B-cell perturbation. Related with this, it was not attainable to optimally time the sampling of lymph node cells in animals belonging to the subgroup of interest. Certain plasma and cellular samples were also unavailable in differing experiments. While partially addressed in in vitro reconstitution, much remains to be investigated for the observed polarity of Env-specific B-cell perturbation in vivo. Similarly, influence of the Nef-G63E phenotype on CD4$^+$ T cells in vivo and its consequence on B-cell modulation remains largely elusive, and in the current work we focused mainly on the molecular properties of the NAb-correlating Nef-G63E mutant strain. HIV/SIV infection uniquely shows a biphasic Tfh dysregulation of acute-phase destruction (*Mattapallil et al., 2005*; *Moukambi et al., 2015*; *Moukambi et al., 2019*) followed by dysregulated chronic-phase hyper-expansion (*Lindqvist et al., 2012*; *Petrovas et al., 2012*; *Cubas et al., 2013*; *Xu et al., 2016*). Further studies are therefore warranted to uncover how the Nef mutant, which is proapoptotic and displays ameliorated PI3K drive, influences Tfh responses at different phases of the infection. Gain-of-function mutations in PI3K-encoding genes appear to affect B cells more strongly (*Asano et al., 2018*). Nonetheless, it is noteworthy that they also evoke basal dysregulation in peripheral CD4$^+$ T cells, which locally recapitulates dysregulation and death of HIV/SIV-infected CD4$^+$ T cells, highlighting the importance of PI3K-mediated signaling in HIV/SIV infection. In any event, we surmise that specific animal models like the current one help to depict a temporal cascade of in vivo events leading to NAb induction, assisting human immunology. As an extension of this study extracting the NAb-involved molecular axis, manipulation/reconstitution experiments shall be designed.

In conclusion, we demonstrated in a non-human primate AIDS model that NAb induction against a difficult-to-neutralize SIV strain occurs after selection of a CD8$^+$ T-cell escape variant with a reduced ability to drive excess PI3K/mTORC2 signaling. These results collectively offer an example of how NAb responses against immunodeficiency viruses can be shaped by both wings of adaptive immune

pressure. Given the key role for Nef-mediated perturbation of PI3K/mTORC2 in NAb resistance, immune cell-intrinsic PI3K/mTORC2 manipulation may offer a new possibility to harness antiviral NAb responses. As human IEIs are increasingly becoming discovered with various immune-perturbing phenotypes and inheritance patterns (*Jauch et al., 2023*), search of other types of analogies between viral immune dysregulations and human IEIs may fuel the discovery of novel targets to modulate and harness immune responses in translational settings.

## Materials and methods
### Materials availability
This study did not generate new unique reagents.

### Rhesus macaques retrospectively utilized for samples
In total, 70 Burmese rhesus macaques (*Macaca mulatta*) (57 males and 13 females) were retrospectively analyzed in this study. Experiments were previously carried out (*Matano et al., 2004*; *Yamamoto et al., 2007*; *Iseda et al., 2016*; *Nomura et al., 2012*; *Ishii et al., 2012*; *Takahashi et al., 2013*; *Nakane et al., 2013*; *Shi et al., 2013*; *Iwamoto et al., 2014*; *Terahara et al., 2014*) at the Tsukuba Primate Research Center, National Institutes of Biomedical Innovation, Health and Nutrition (NIBIOHN), with the help of the Corporation for Production and Research of Laboratory Primates and the Institute for Frontier Life and Medical Sciences, Kyoto University (IFLMS-KU) after approval by the Committee on the Ethics of Animal Experiments of NIBIOHN and IFLMS-KU under the guidelines for animal experiments at NIBIOHN, IFLMS-KU, and National Institute of Infectious Diseases in accordance with the 'Guidelines for Proper Conduct of Animal Experiments' established by Science Council of Japan. The experiments were in accordance with the 'Weatherall report on the use of non-human primates in research' recommendations. Animals were housed in adjoining individual primate cages, allowing them to make sight and sound contact with one another for social interactions, where the temperature was kept at 25°C with light for 12 hours per day. Animals were fed with apples and commercial monkey diet (Type CMK-2, Clea Japan, Inc). Blood collection and virus challenge were performed under ketamine anesthesia.

### Cells and viruses
MT4-R5 cells were maintained in RPMI1640 (Invitrogen) supplemented with 10% fetal bovine serum (FBS) (Clontech) and antibiotics. HSC-F cells (cynomolgus $CD4^+$ T-cell line) and HSR5.4 cells (rhesus macaque $CD4^+$ T-cell line) were maintained in RPMI1640 (Invitrogen) supplemented with 10% FBS (Clontech), human IL-2 (10 IU/ml, Roche), 4-(2-hydroxyethyl)-1-piperazineethanesulfonic acid (HEPES) (Invitrogen), and 2-mercaptoethanol (Gibco). $SIV_{mac239}$ molecular clone derivative was generated by mutagenesis PCR (Agilent Technologies) using the primers listed. For virus preparation, COS-1 cells were transfected with $pBR_{mac239}$ proviral DNA using FuGENE 6 (Promega). At 48 hours post-transfection, culture supernatants were harvested, centrifuged, and filtered through a 0.45 μm pore-size filter (Merck Millipore). To titrate infectivity, prepared viruses were serially diluted and infected on HSC-F cells in 96-well plates (Falcon) in quadruplicate. At 10 days p.i., the endpoint was determined using SIV p27 antigen ELISA kit (ABL), and virus infectivity was calculated as the 50% tissue culture infective dose ($TCID_{50}$) according to the Reed–Muench method.

### Identification of $SIV_{mac239}$-NAb inducers
Burmese rhesus macaques previously challenged with the highly pathogenic molecular clone virus $SIV_{mac239}$ (n = 70) were retrospectively examined approximately up to 2 years for their plasma NAb profiles. Animals were challenged intravenously with 1000 $TCID_{50}$ of $SIV_{mac239}$. In the current study, virological and immunological profiles were compared between the newly identified NAb inducers (n = 9) and representative NAb non-inducers (n = 19). Sex distribution of the NAb inducers (eight males and one female) did not significantly differ with the total non-inducers (49 males and 12 females, p=0.99 by Fisher's exact test). These NAb non-inducers and eight NAb inducers except R06-032 were previously partially reported for their plasma viral loads (*Yamamoto et al., 2007*; *Iseda et al., 2016*; *Nomura et al., 2012*; *Takahashi et al., 2013*; *Nakane et al., 2013*; *Iwamoto et al., 2014*; *Figure 1B*). MHC-I haplotypes and alleles were determined by reference strand-mediated conformation analysis,

PCR-SSP (PCR amplification utilizing sequence-specific priming), and cloning as described (*Naruse et al., 2010*). MHC-I binding prediction was made on April 27, 2013, using the IEDB analysis resource NetMHCpan tool (*Hoof et al., 2009*). Alleles of interest in the study have been previously identified in macaques (*Nomura et al., 2012*; *Evans et al., 1999*; *Sette et al., 2012*). Macaques R06-032, R03-015, R01-010, and R05-010 received a prime-boost vaccination (*Matano et al., 2004*) composed of a DNA prime/intranasal Sendai virus vector expressing SIV$_{mac239}$ Gag (SeV-Gag). R03-015, R06-019, R06-038, and R10-001 received 300 mg of non-specific rhesus IgG at day 7 post-SIV$_{mac239}$ challenge as an experimental control in our previous reports (*Yamamoto et al., 2007*; *Iseda et al., 2016*; *Nakane et al., 2013*).

## Plasma viral load quantitation

Plasma viral RNA samples were extracted with High Pure Viral RNA kit (Roche Diagnostics). Serial five-fold sample dilutions were amplified in quadruplicate by reverse transcription and nested PCR using SIV$_{mac239}$ *gag*-specific primers to determine end point via the Reed–Muench method as described previously (*Matano et al., 2004*; *Iseda et al., 2016*). The lower limit of detection is approximately 400 viral RNA copies/ml plasma. Viral loads have been previously partially reported (*Yamamoto et al., 2007*; *Nomura et al., 2012*; *Takahashi et al., 2013*; *Nakane et al., 2013*; *Iwamoto et al., 2014*).

## SIV$_{mac239}$-specific neutralization assay

NAbs were titrated as described (*Yamamoto et al., 2007*; *Iseda et al., 2016*). Serial twofold dilutions of heat-inactivated plasma or polyclonal IgG affinity-purified with Protein G Sepharose 4 Fast Flow (GE Healthcare) from heat-inactivated and filtered plasma were mixed with 10 TCID$_{50}$ of SIV$_{mac239}$ at a 1:1 ratio (5 µl:5 µl) in quadruplicate. After 45-minute incubation at room temperature, the 10 µl mixtures were added into 5 × 10$^4$ MT4-R5 cells/well in 96-well plates. Progeny virus production in day 12 culture supernatants was examined by SIV p27 ELISA (ABL) to determine 100% neutralizing endpoint. The lower limit of titration is 1:2. Neutralization in three out of four wells at a dilution of 1:2 is defined as marginally NAb-positive (<1:2). Results were comparable when the same assay was performed with macaque HSC-F cells (*Akari et al., 1996*) as targets. NAb inducers R01-012, R02-004, R01-010, R02-007, and NAb non-inducer R01-011 were previously partially reported for their NAb titers measured with the same method using MT4 cells as targets (*Kawada et al., 2007*), which derived comparable results. For the assessment of neutralizing activity in IgG, SIV$_{mac239}$-specific IgGs purified from pools of plasma with SIV$_{mac239}$-specific NAb titers were obtained from each animal as described (*Yamamoto et al., 2007*). After complement heat inactivation at 56°C, 30 minutes, and 0.45 µm filtration, IgGs were purified by Protein G Sepharose 4 Fast Flow (GE Healthcare) and concentrated by Amicon Ultra 4, MW 50,000 (Millipore) to 30 mg/ml and similarly examined for their 10 TCID$_{50}$ SIV$_{mac239}$ killing titers on MT4-R5 cells.

## SIV Env-specific IgG ELISA and immunoblotting

Plasma Env-specific IgG titers were measured as described (*Nakane et al., 2013*). SIV$_{mac251}$ Env gp120 (ImmunoDiagnostics) were coated on 96-well assay plates (BD) at 1000 ng/ml (100 µl/well). Wells were prewashed with phosphate-buffered saline (PBS), blocked with 0.5% bovine serum albumin (BSA)/PBS overnight, and plasma samples were incubated at a 1:20 dilution (5 µl:95 µl) for 2 hours. Wells were washed with PBS and SIV Env-bound antibodies were detected with a horseradish peroxidase (HRP)-conjugated goat anti-monkey IgG (H+L) (Bethyl Laboratory) and SureBlue TMB 1-Component Microwell Peroxidase Substrate (KPL). Absorbance at 450 nm was measured. Samples from week 0 pre-challenge and month 3, month 6, and year 1 post-challenge were assessed in duplicate. Week 0 average values were subtracted from the corresponding later time point values for calibration. For immunoblotting, SIV virion-specific IgGs in plasma were detected with a SIV$_{mac239}$-cross-reactive western blotting system (ZeptoMetrix). In the NAb non-inducers, samples from those close to rapid progression (euthanized due to AIDS progression within approximately 1 year) known for low plasma anti-SIV reactivity (*Hirsch et al., 2004*; *Nakane et al., 2013*) were not included.

## Sequencing

Sequencing was performed as described (*Matano et al., 2004*; *Iseda et al., 2016*). Viral cDNA fragments spanning from nt (nucleotide) 4829 to nt 7000, nt 6843 to nt 8831, and nt 8677 to nt 10196

in SIV$_{mac239}$ (GenBank accession number MM33262) covering SIV *env* and *nef* were amplified from plasma viral RNA by nested RT-PCR using Prime-Script one-step RT-PCR kit v2 (TaKaRa) and KOD-Plus v2 (Toyobo). PCR products were either directly sequenced or subcloned with a TOP10-transforming TOPO blunt-end cloning system (Invitrogen). Sequencing was performed using dye terminator chemistry with an ABI 3730 DNA sequencer (Applied Biosystems). On average, 15 clones were obtained per sample and 20 clones were assessed when *nef* mutations of interest in early time points were subdominant.

## SIV$_{mac239}$-specific CD8$^+$ T-cell responses

Virus-specific CD8$^+$ T-cell frequencies were measured as described (*Matano et al., 2004*; *Iseda et al., 2016*). Peripheral blood mononuclear cells (PBMCs) were cocultured for 6 hours with autologous herpesvirus papio-immortalized B lymphoblastoid cell lines (B-LCLs) pulsed with Nef peptides (Sigma-Aldrich Japan) at 1 μM concentration or as indicated otherwise under GolgiStop (monensin, BD) presence. Intracellular gamma interferon (IFN-γ) staining was performed using Cytofix/Cytoperm kit (BD) and the following conjugated anti-human monoclonal antibodies (mAbs): anti-CD4-FITC (M-T477, BD Pharmingen), anti-CD8-PerCP (SK1, BD Biosciences), anti-CD3-APC (SP34-2, BD Pharmingen), and anti-IFN-γ-PE (4S.B3, BioLegend). Specific CD8$^+$ T-cell frequencies were determined by subtracting nonspecific IFN-γ$^+$ CD8$^+$ T-cell frequencies from those after peptide-specific stimulation; frequencies beneath 100 cells/million PBMCs were considered negative. Cells acquired by FACS Canto II (BD) were analyzed using FACS Diva (BD) and FlowJo (Treestar). Approximately $1 \times 10^5$ PBMCs were gated per test.

## Nef-mediated signaling perturbation analysis

Virus supernatants obtained from COS-1 cells after transfection with WT or mutant SIV$_{mac239}$ molecular clones were used for infection of CD4$^+$ T-cell lines, cynomolgus macaque-derived HSC-F, and rhesus macaque-derived HSR5.4 (*Akari et al., 1996*). QuikChange II XL site-directed mutagenesis kit (Agilent Technologies) was used to construct mutant SIV$_{mac239}$ molecular clones possessing *nef* mutations Nef-G63E (G-to-A mutation at nt 9520) and Nef-G63R (G-to-A mutation at nt 9519) from the WT SIV$_{mac239}$ molecular clone (*Kestler et al., 1991*) (nt number from GenBank accession number M33262). Cells ($1 \times 10^5$ cells/well in U-bottomed 96-well culture plates [BD]) were infected with WT or mutant SIV$_{mac239}$ cultured in RPMI-1640 medium supplemented with 10% FBS for indicated periods at MOI 0.1 (intracellular signaling analysis) or 0.001 (supernatant analysis). Culture supernatants were subjected to measurement of SIV capsid p27 concentrations by ELISA. Harvested cells were fixed and permeabilized with Cytofix/Cytoperm kit, washed twice, and subjected to immunostaining. The following antibodies were used; anti-SIV$_{mac251}$ Gag p27 mAb (ABL) manually conjugated with Alexa 647 (Life Technologies), anti-SIV$_{mac251}$ Nef mAb (clone 17, epitope peptide corresponding to SIV$_{mac239}$ Nef 71–80 not including the residue G63: Thermo Fisher Scientific/Pierce) manually conjugated with Alexa 488 or Alexa 647, anti-human CD3-APC-Cy7 (SP34-2, BD Pharmingen), anti-human CD4-PerCP (L200, BD Pharmingen), anti-human HLA-ABC-FITC (G46-2.6, BD Pharmingen), anti-human CXCR4-APC (12G5, BioLegend), BST2-Alexa 647 (RS38E, BioLegend), Alexa 488-conjugated anti-phospho-Akt (Ser473) (D9E, CST), and Alexa 488-conjugated anti-phospho-Akt (Thr308) (C31E5E, CST). Cells acquired by FACS Canto II were analyzed using FACS Diva and FlowJo. Approximately $5 \times 10^4$ cells were gated for each test.

## Ferritin ELISA

Plasma ferritin levels in NAb inducers and control animals were analyzed using monkey ferritin sandwich ELISA kit (LS Bio) according to the manufacturer's instructions.

## PI3K stimulation assay

$5 \times 10^4$ HSC-F cells were infected with WT or Nef-G63E mutant SIV$_{mac239}$ at MOI 0.2 and cultured for 1 day in medium supplemented with 10% (normal), 1% (1/10 starvation), or 0.1% (1/100 starvation) FBS. For ligand stimulation (*Figure 4—figure supplement 1C*), cells at the end of 1-day culture were pulsed for 20 minutes with 40 ng/ml of recombinant human IFN-γ (Gibco/Thermo Fisher Scientific), 100 IU/ml of recombinant human IL-2 (Roche Diagnostics), or 10 μg/ml of SIV$_{mac251}$ Env gp130 (ImmunoDx). Cells were intracellularly stained using Cytofix/Cytoperm kit with anti-SIV$_{mac251}$ Nef mAb or

anti-SIV$_{mac251}$ p27 mAb manually conjugated to Alexa 647 and PE-conjugated anti-human phospho-Akt (Ser473) (D9E, CST) or Alexa 488-conjugated anti-human phospho-Akt (Ser473). Cells acquired by FACS Canto II were analyzed using FACS Diva and FlowJo. Approximately $7 \times 10^4$ HSC-F cells were gated per test.

## Transcriptome analysis

Total RNAs were extracted using RNeasy Plus Mini kit (QIAGEN) from $2 \times 10^6$ HSC-F cells 1 day after infection with WT or Nef-G63E mutant SIV$_{mac239}$ at MOI 5. Negative control RNA samples were extracted from $2 \times 10^6$ uninfected HSC-F cells after culture with the same condition. Three sets of experiments were performed. Total RNA samples were subjected to a quality control (QC) analysis using an Agilent 2100 Bioanalyzer. The obtained amounts of total RNAs were 12.37 ± 0.39 (uninfected), 8.80 ± 0.44 (WT), and 9.02 ± 0.17 (Nef-G63E) µg (p=0.66 for WT vs. Nef-G63E by unpaired $t$-test). In all samples, two bands of 18S and 28S rRNA were confirmed and the RNA integrity number (RIN) was 10. 500 ng of total RNA samples were processed with GeneChip WT Plus reagent (Affymetrix/Thermo Fisher Scientific) to produce 150 µl of fragmented and labeled cDNA samples. These were incubated with a Human Gene 2.0 ST Array (Affymetrix/Thermo Fisher Scientific) for 16 hours, 60 rpm at 45°C using GeneChip hybridization oven 645 (Affymetrix/Thermo Fisher Scientific). Results were scanned with GeneChip Scanner 3000 7G (Affymetrix/Thermo Fisher Scientific) and processed with Affymetrix Expression Console Software (Affymetrix/Thermo Fisher Scientific) according to the manufacturer's instructions. Expression values were normalized by the RMA method. Genes above mean background expression within the cognate sample and showing significant difference between WT and Nef-G63E SIV (p<0.05 via unpaired $t$-test for log$_2$-transformed values, 768 candidates) were determined. Akt-related genes exhibiting a change of approximately 10% or more were representatively extracted and manually aligned by the authors.

## Peripheral CD4$^+$ T-cell surface staining

Cryopreserved/thawed PBMCs were stained for 30 minutes at 4°C with the following reagents or conjugated anti-human mAbs: Live/Dead Aqua (Life Technologies), anti-CD4-PerCP (L200, BD Pharmingen), anti-CD8-APC-Cy7 (RPA-T8, BD Pharmingen), anti-CD3-Alexa 700 (SP34-2, BD Pharmingen), anti-CD95-PE-Cy7 (DX2, eBioscience), anti-CXCR5-PE (87.1, eBioscience), anti-PD-1-Brilliant Violet 421 (EH12.2H7, BioLegend), and anti-CXCR3-Alexa 488 (G025H7, BioLegend). Cells acquired by FACS LSRII Fortessa (BD) were analyzed using FACS Diva and FlowJo. Approximately $1.5 \times 10^5$ PBMCs were gated per test.

## Proximity ligation assay

A flow cytometry-based arrangement of PLA (*Avin et al., 2017*) was performed to quantitatively assess Nef binding to candidate interacting molecules. $1 \times 10^5$ HSC-F cells were infected with SIV$_{mac239}$ at MOI 0.05 in U-bottomed plates and permeated with Cytofix/Perm kit (BD Biosciences). After two washes, they were resuspended in 0.5% BSA/PBS for the prevention of experimental procedure-related loss and stained with mouse anti-SIV$_{mac251}$ Nef mAb (clone 17, Thermo Fisher Scientific/Pierce) or mouse IgG1 isotype control mAb (P3.6.2.8.1, Abcam) in combination with either of the following rabbit antibodies: anti-GβL (86B8, CST), anti-mTOR (7C10, CST), polyclonal anti-human PI3K p85 (Merck Millipore), anti-PI3 Kinase p110α (C73F8, CST), or rabbit IgG1 isotype control mAb (DA1E, CST). Antibody-stained cells were subsequently probed with Duolink In Situ PLA Probe anti-mouse PLUS and anti-rabbit MINUS probes (Sigma/Merck). Next, they were detected for intermolecular binding using Duolink flow PLA detection kit (Green) (Sigma/Merck) with a reaction time of 100 minutes for post-mouse/rabbit probe linking amplification. Finally, these PLA-subjected cells were additionally stained with anti-SIV$_{mac251}$ Gag p27 mAb manually conjugated with Alexa 647 and anti-CD4-PerCP for 20 minutes at 4°C. Cells acquired by FACS Canto II were analyzed using FACS Diva and FlowJo. Approximately $1 \times 10^5$ cells were gated per test. Binding index (Y-axis) was calculated by deriving the deviation from the summation of: {baseline P: (MFI of background reaction with mouse isotype control/rabbit isotype control) + anti-Nef antibody-derived background Q: [(MFI of reaction with mouse anti-Nef/rabbit isotype control) – (MFI of background reaction with mouse isotype control/rabbit isotype control)]+anti-binding partner antibody-derived background R:[(MFI of reaction

with mouse isotype control/rabbit anti-binding partner molecule) – (MFI of background reaction with mouse isotype control/rabbit isotype control)]}.

## Co-immunoprecipitation analysis

$1 \times 10^5$ HSC-F cells were infected with $SIV_{mac239}$ at MOI 0.05 in U-bottomed plates in quadruplicate for each SIV strain for 3 days and pooled for each for acquiring cell pellets. These pellets were lysed with Capturem IP & Co-IP Kit lysis buffer (Takara), and portions for each were immunoprecipitated by anti-Sin1 mAb (D7G1A, CST) (1:50 dilution, 20 minutes, room temperature). Spin column membrane-bound immunoprecipitates were obtained by centrifugation with Capturem IP & Co-IP Kit (Clontech/Takara). Whole-cell lysates and Sin1-immunoprecipitated products were subjected to SDS-polyacrylamide gel electrophoresis separation on a Mini Protean TGX 4–15% gel (Bio-Rad) and transferred to a Immun-Blot PVDF membrane (Bio-Rad). Immunoblotting was performed by mouse anti-$SIV_{mac251}$ Nef mAb (clone 17, Thermo Fisher Scientific/Pierce) primary antibody probing (1:1000 dilution, 18 hours, 4°C) and Mouse TrueBlot ULTRA anti-mouse Ig HRP (eB144, Rockland Immunochemicals) secondary antibody incubation (1:1000 dilution, 30 minutes, room temperature). Nef-specific bands in infected controls were visualized by enhanced chemiluminescence using SuperSignal West Pico PLUS (Thermo Fisher Scientific). Bands auto-detectable using Image Lab software (Bio-Rad) were analyzed for signal intensities. Uncropped images are provided as supplementary data.

## Cell death assay

Infected cell apoptosis was measured by methods previously described (*Jauch et al., 2023*). $1 \times 10^5$ HSC-F cells were infected with $SIV_{mac239}$ at MOI 0.1 in U-bottomed plates (six wells/control) for each SIV strain for 3 days. These cells were stained with CD4-FITC (M-T477, BD Pharmingen) for 15 minutes at room temperature, washed once, and next stained with APC Annexin V (BioLegend) in Annexin V Binding Buffer (BioLegend) for 20 minutes at room temperature. Stained reactions were then diluted fivefold with Annexin V Binding Buffer and subjected to analysis. Cells acquired by FACS Lyric were analyzed using FACS Suite and FlowJo. Approximately $1 \times 10^4$ HSC-F cells were gated per test.

## Quantitative in vivo imaging flow cytometry

Cryopreserved/thawed lymph node cells (LNCs) from several persistently SIV-infected macaques used in previous experiments (*Nakane et al., 2013*; *Takahashi et al., 2013*; *Iwamoto et al., 2014*) were seeded in V-bottomed 96-well plates (Nunc) and blocked with 25 µg/ml of anti-human CD4 (clone L200: BD) in 100 µl volume for 15 minutes at 4°C. After three washes, they were stained with 10 µg/ml of recombinant $SIV_{mac239}$ Env gp140-biotin (Immune Technology) for 30 minutes at 4°C. Cells were then stained with anti-CD19-PC5.5 (J3-119, Beckman Coulter), Live/Dead Aqua (Invitrogen), and streptavidin-APC-Cy7 (BioLegend) for 30 minutes at 4°C. After two washes, cells were processed with Cytofix/Cytoperm kit (BD) for 20 minutes at 4°C, washed twice, and next intracellularly stained with PE-conjugated anti-human phospho-Akt (Ser473) (D9E, CST) and anti-$SIV_{mac251}$ Nef mAb (clone 17, Thermo Fisher Scientific/Pierce) manually conjugated with Alexa 488 or mouse IgG1 isotype control mAb (P3.6.2.8.1, Abcam) for 30 minutes at 4°C. After final two washes, cells were suspended in 0.8% PFA/PBS. All centrifugations for washing ($1200 \times g$, 2 minutes) were performed at 4°C. Cells were subjected to image acquisition with Image Stream X MKII imaging flow cytometer (Amnis/Merck Millipore/Luminex) and analyzed with IDEAS 6.3 software (Amnis/Merck Millipore/Luminex). Approximately $5 \times 10^5$ LNCs were acquired for analyses (*Figure 5*). A custom-implemented linear discriminant analysis-based machine learning module (Luminex) was utilized for verifying candidate Nef staining signal-related secondary parameters (*Figure 5*, *Figure 5—figure supplement 1*) for their efficacy of target population separation.

## B-cell Nef invasion in vitro assay

For Nef invasion assay, $5 \times 10^4$ HSC-F cells were infected at MOI 1 for 48 hours. These were then additionally cocultured with $7.5 \times 10^4$ Ramos B cells for 12 hours with or without addition of macaque T cell-stimulating anti-CD3 antibody (FN-18, Abcam) at 1 µg/ml. Cells were double-stained with Alexa 488-conjugated mouse anti-Nef (clone 17: Thermo Scientific/Pierce) and anti-CD19-PC-5.5 (J3-119, Beckman Coulter). For graphical visualization (*Figure 6A*), interacting Nef+ infected cell-invaded B cell doublet images acquired with ImageStream X MKII (Amnis/Merck Millipore) were analyzed using

IDEAS 6.3 (Amnis/Merck Millipore/Luminex) with the defined noise-cancelling gating (via gating Nef pixel signal contrast$^{lo}$-Nef mean pixel per object$^{hi}$ cells). For Nef invasion modulation verification on conventional flow cytometry, the aforementioned culture condition was similarly used. Anti-CD3 antibody (FN-18, Abcam) addition at 1 μg/ml and phorbol 12-myristate 13-acetate (PMA, Sigma) plus ionomycin (Sigma) addition (10 ng/ml and 50 ng/ml, respectively) were evaluated. Cells were stained with Alexa 647-conjugated mouse anti-Nef (clone 17: Thermo Scientific/Pierce), anti-CD19-PC-5.5 (J3-119, Beckman Coulter), anti-human phospho-Akt (Ser473)-Alexa 488 (D9E, CST), and Live/Dead Near-IR (Invitrogen) for dead cell exclusion. Dead cell frequencies were less than 3%. For viral phenotype evaluation, $1 \times 10^5$ HSC-F cells (surface CD40L$^+$) were pre-infected at MOI 0.2 with WT or Nef-G63E mutant SIV$_{mac239}$ 6 hours before coculture. $1.5 \times 10^5$ CD20$^+$ B cells, positively selected from fresh PBMCs of uninfected rhesus macaques via anti-CD20 microbeads (Miltenyi Biotec), were cocultured with these HSC-F cells at an E:T ratio of 2:3 for 3 days in the presence of 40 ng/ml carrier-free recombinant human IL-4 (R&D Systems). Cells were surface-stained with anti-human CD20-PerCP (2H7, BioLegend) and intracellularly stained using Cytofix/Cytoperm kit with anti-SIV$_{mac251}$ Nef mAb manually conjugated to Alexa 488 and Alexa 647-conjugated anti-human phospho-Akt (Ser473) (D9E, CST). For *Figure 6B and C*, cells acquired by FACS Canto II were analyzed using FACS Diva and FlowJo. Approximately $5 \times 10^4$ CD20$^+$ primary B cells and $3 \times 10^4$ CD19$^+$ Ramos B cells were analyzed.

## Peripheral SIV Env-specific B-cell responses

Peripheral SIV Env gp140-specific memory B cells (B$_{mem}$) and PBs were measured by flow cytometry with procedures modified from previous reports and our experience (*Silveira et al., 2015*; *Hau et al., 2022*). First, cryopreserved/thawed PBMCs seeded in V-bottomed 96-well plates (Nunc) were blocked with 25 μg/ml of anti-human CD4 (clone L200: BD) in 100 μl volume for 15 minutes at 4°C. This concentration attains blockade of promiscuous SIV Env binding to CD4 comparable to levels by magnetic depletion of CD3$^+$ T cells (data not shown). After three washes, they were next stained with 10 μg/ml of recombinant SIV$_{mac239}$ Env gp140-biotin (Immune Technology) for 30 minutes at 4°C. They were subsequently stained with the following anti-human antibodies and fluorochromes for 30 minutes at 4°C with combinations as follows; PB/B$_{mem}$: anti-CD3-APC-Cy7 (SP34-2, BD Pharmingen), anti-CD8-APC-Cy7 (RPA-T8, BD Pharmingen), anti-CD14-APC-Cy7 (M5E2, BioLegend), anti-CD16-APC-Cy7 (3G8, BioLegend), streptavidin-Brilliant Violet 421 (BioLegend), anti-CD19-PC5.5 (J3-119, Beckman Coulter) and Live/Dead Aqua (Invitrogen); PB: anti-CD10-APC-Cy7 (HI10a, BioLegend), anti-CD11c-PE-Cy7 (3.9, BioLegend), anti-CD123-PE-Cy7 (6H6, BioLegend), and anti-HLA-DR-PE-Texas Red (TU36, Invitrogen); B$_{mem}$: anti-CD27-PE/Dazzle 594 (M-T271, BioLegend), anti-CD10-PE-Cy7 (HI10a, BioLegend), anti-IgD-FITC (DaKo), anti-CD38-Alexa 647 (AT1, Santa Cruz Biotechnology), anti-IgG-Alexa 700 (G18-145, BD Pharmingen), and anti-CD138-PE (DL-101, eBioscience). After two washes, B$_{mem}$ samples were suspended in 0.8% PFA/PBS. For PB staining, surface-stained cells were further processed with Cytofix/Cytoperm kit (BD) for 20 minutes at 4°C, washed twice, and next intracellularly stained with 10 μg/ml of recombinant SIV$_{mac239}$ Env gp140-biotin (Immune Technology) for 30 minutes at 4°C. After three washes, they were next stained with anti-human IgG-PE (G18-145, BD), anti-human/mouse IRF4-eFluor 660 (3E4, eBioscience), anti-human Ki67-Alexa 700 (B56, BD Pharmingen), and streptavidin-Alexa 488 (BioLegend) for 30 minutes at 4°C. After final two washes, cells were suspended in 0.8% PFA/PBS. Cells acquired with FACS LSRII Fortessa were analyzed with FACS Diva and FlowJo. Approximately $1.5 \times 10^5$ PBMCs were acquired for B$_{mem}$ and $4 \times 10^5$ PBMCs were acquired for PB analyses (*Figure 6*). All centrifugations for washing ($1200 \times g$, 2 minutes) were performed at 4°C. Env-specific PB frequencies were 0 cells/million PBMCs for pre-challenge samples in all animals examined, ruling out background confounding for high-sensitivity quantitation of the population.

## Statistical analysis

Analyses were performed using Prism 8 (GraphPad Software). $p < 0.05$ was considered significant in two-tailed unpaired *t*-tests, paired *t*-tests, Mann–Whitney *U* tests, Fisher's exact tests, Wilcoxon signed-rank tests, one-way ANOVA with Tukey's post hoc multiple comparison tests and two-way ANOVA with Sidak's post hoc multiple comparison tests. For analysis of Nef-invaded B-cell pAkt levels, median fluorescence intensities (mFIs) were analyzed due to relatively small numbers of Nef$^+$ B cells. Analyses involving pAkt Ser473 levels all derived comparable results between mFI and MFI for

WT versus Nef-G63E virus. Machine learning-based rating of Nef signal parameters was verified with IDEAS 6.3 Machine Learning Module (Luminex).

## Acknowledgements

We thank Fumiko Ono, Koji Hanari, Katsuhiko Komatsuzaki, Akio Hiyaoka, Hiromi Ogawa, Keiko Oto, Hirofumi Akari, Yasuhiro Yasutomi, Hiromi Sakawaki, Tomoyuki Miura, and Yoshio Koyanagi for animal experiment assistance; Hiroyuki Kogure, Sayuri Seki, and Julia R Hirsiger for imaging cytometry technical assistance; and Masako Nishizawa, Trang Thi Thu Hau, Shigeyoshi Harada, Takushi Nomura, Akiko Takeda, Taku Nakane, Nami Iwamoto, Taeko K Naruse, Akinori Kimura, and Mark S de Souza for their help. HY thanks Shoi Shi, Bettina Stolp, Jens V Stein, Makoto Yamagishi, and Yusuke Yanagi for conceptual discussions, and Jun Abe and Mike Recher for collaborative support.

## Additional information

### Funding

| Funder | Grant reference number | Author |
|---|---|---|
| Japan Agency for Medical Research and Development | JP24fk0410066 | Hiroyuki Yamamoto |
| Japan Agency for Medical Research and Development | JP21jk0210002 | Tetsuro Matano |
| Ministry of Education, Culture, Sports, Science and Technology | 24K21287 | Hiroyuki Yamamoto |
| Ministry of Education, Culture, Sports, Science and Technology | 21H02745 | Tetsuro Matano |
| Takeda Science Foundation | | Hiroyuki Yamamoto |
| Imai Memorial Trust for AIDS Research | | Hiroyuki Yamamoto |
| Mitsui Sumitomo Insurance Welfare Foundation | | Hiroyuki Yamamoto |
| Japan Agency for Medical Research and Development | JP22wm0325006 | Hiroyuki Yamamoto |
| Japan Agency for Medical Research and Development | JP19fm0208017 | Hiroyuki Yamamoto |
| Japan Agency for Medical Research and Development | JP20fk0410022 | Hiroyuki Yamamoto |
| Japan Agency for Medical Research and Development | JP18fk0410003 | Tetsuro Matano |
| Japan Agency for Medical Research and Development | JP20fk0410011 | Tetsuro Matano |
| Japan Agency for Medical Research and Development | JP20fk0108125 | Tetsuro Matano |

| Funder | Grant reference number | Author |
|---|---|---|
| Japan Agency for Medical Research and Development | JP20jm0110012 | Tetsuro Matano |
| Japan Agency for Medical Research and Development | JP21fk0410035 | Tetsuro Matano |
| Ministry of Education, Culture, Sports, Science and Technology | 17H02185 | Hiroyuki Yamamoto |
| Ministry of Education, Culture, Sports, Science and Technology | 18K07157 | Hiroyuki Yamamoto |

The funders had no role in study design, data collection and interpretation, or the decision to submit the work for publication.

## Author contributions

Hiroyuki Yamamoto, Conceptualization, Resources, Data curation, Formal analysis, Supervision, Funding acquisition, Validation, Investigation, Visualization, Methodology, Writing – original draft, Project administration, Writing – review and editing; Tetsuro Matano, Resources, Supervision, Funding acquisition, Writing – review and editing

## Author ORCIDs

Hiroyuki Yamamoto (iD) https://orcid.org/0000-0002-0708-9373
Tetsuro Matano (iD) https://orcid.org/0000-0003-3096-6749

Reviewer #3 (Public review): https://doi.org/10.7554/eLife.88849.4.sa1
Author response https://doi.org/10.7554/eLife.88849.4.sa2

# Additional files

## Supplementary files

MDAR checklist

## Data availability

Data were analyzed using existing computational packages. The accession number for the transcriptome data [WT SIVmac239-infected HSC-F cell line (n = 3), Nef-G63E SIVmac239-infected HSC-F cell line (n = 3) and uninfected HSC-F cell line (n = 3)] has been deposited under the accession number GenBank: GSE65806.

The following dataset was generated:

| Author(s) | Year | Dataset title | Dataset URL | Database and Identifier |
|---|---|---|---|---|
| Yamamoto H | 2016 | Gene expression profile in CD4+ T-cell infection with a SIV mutant related to altered humoral immune responses | https://www.ncbi.nlm.nih.gov/geo/query/acc.cgi?acc=GSE65806 | NCBI Gene Expression Omnibus, GSE65806 |

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
