## [Editor Report · eLife Assessment]

Yamamoto and Matano provide **convincing** evidence that a G63E/R CD8+ T-cell escape mutation in the accessory viral protein Nef promote the induction of neutralizing antibody (nAb) responses in rhesus macaques infected with SIVmac239, which is usually largely resistant to neutralization. Functional analyses support that this mutation specifically impairs Nef’s ability to stimulate PI3K/Akt/mTORC2 signaling. This **important** study suggests that the accessory viral protein Nef impairs B cell function and effective humoral immune responses and is of interest for researchers and physicians interested in HIV/AIDS and vaccine development.

---

## [Referee Report · Reviewer #3 (Public review)]

Human and simian immunodeficiency viruses (HIV and SIV, respectively) evolved numerous mechanisms to compromise effective immune responses but the underlying mechanisms remain incompletely understood. Here, Yamamoto and Matano examined the humoral immune response in a large number of rhesus macaques infected with the difficult-to-neutralize SIVmac239 strain and identified a subgroup of animals showing significant neutralizing Ab responses. Sequence analyses revealed that in most of these animals (7/9) but only a minority in the control group (2/19) SIVmac variants containing a CD8+ T-cell escape mutation of G63E/R in the viral Nef gene emerged. Functional analyses revealed that this change attenuates the ability of Nef to stimulate PI3K/Akt/mTORC2 signalling. The authors propose that this improved induction of SIVmac239 nAb is reciprocal to antibody dysregulation caused by a previously identified human PI3K gain-of-function mutation associated with impaired anti-viral B-cell responses. Altogether, the results suggest that PI3K signalling plays a role in B-cell maturation and generation of effective nAb responses. Preliminary data indicate that Nef might be transferred from infected T cells to B cells by direct contact. However, the exact mechanism and the relevance for vaccine development requires further studies

The strengths of the study are that the authors analyzed a large number of SIVmac-infected macaques to unravel the biological significance of the known effect of the interaction of Nef with PI3K/Akt/mTORC2 signaling. This is interesting and may provide a novel means to improve humoral immune responses to HIV. In the revised version the authors made an effort to address previous concerns. Especially, they provide data supporting that Nef might be transferred to B cells by direct cell-cell contact. In addition, they provide some evidence that G63R that also emerged in most animals does not share the disruptive effect of G63G although experimental examination and discussion why G63R might emerge remains poor. A weakness that remains is that some effects of the G63E mutation are modest and effects were not compared to SIVmac constructs lacking Nef entirely. The evidence for a role of Nef G63E mutation on PI3K and the association with improved nAb responses is convincing and it is appreciated that the authors provide additional evidence for a potential impact of "soluble" Nef on neighboring B cells. The presentation of the experimental set-up and the results has been improved but is in part still challenging to comprehend. It seems that direct cell-cell contact is required and membranes are exchanged. Since Nef is associated with cellular membranes this might lead to some transfer of Nef to B cells. However, the immunological and functional consequences of this largely remain to be determined. Alternatively, Nef-mediated manipulation of helper CD4 T cells might also impact B cell function and effective humoral immune responses. Additional editing of the manuscript has been performed to make the results accessible to a broad readership.

---

## [Author Response]

The following is the authors’ response to the previous reviews.

**Reviewer #3 (Public review):**
Human and simian immunodeficiency viruses (HIV and SIV, respectively) evolved numerous mechanisms to compromise effective immune responses but the underlying mechanisms remain incompletely understood. Here, Yamamoto and Matano examined the humoral immune response in a large number of rhesus macaques infected with the difficult-to-neutralize SIVmac239 strain and identified a subgroup of animals showing significant neutralizing Ab responses. Sequence analyses revealed that in most of these animals (7/9) but only a minority in the control group (2/19) SIVmac variants containing a CD8+ T-cell escape mutation of G63E/R in the viral Nef gene emerged. Functional analyses revealed that this change attenuates the ability of Nef to stimulate PI3K/Akt/mTORC2 signalling. The authors propose that this improved induction of SIVmac239 nAb is reciprocal to antibody dysregulation caused by a previously identified human PI3K gain-of-function mutation associated with impaired anti-viral B-cell responses. Altogether, the results suggest that PI3K signalling plays a role in B-cell maturation and generation of effective nAb responses. Preliminary data indicate that Nef might be transferred from infected T cells to B cells by direct contact. However, the exact mechanism and the relevance for vaccine development requires further studiesStrengths of the study are that the authors analyzed a large number of SIVmac-infected macaques to unravel the biological significance of the known effect of the interaction of Nef with PI3K/Akt/mTORC2 signaling. This is interesting and may provide a novel means to improve humoral immune responses to HIV. In the revised version the authors made an effort to address previous concerns. Especially, they provide data supporting that Nef might be transferred to B cells by direct cell-cell contact. In addition, the provide some evidence that G63R that also emerged in most animals does not share the disruptive effect of G63G although experimental examination and discussion why G63R might emerge remains poor. Another weakness that remains is that some effects of the G63E mutation are modest and effects were not compared to SIVmac constructs lacking Nef entirely. The evidence for a role of Nef G63E mutation on PI3K and the association with improved nAb responses was largely convincing and it is appreciated that the authors provide additional evidence for a potential impact of "soluble" Nef on neighboring B cells. However, the experimental set-up and the results are difficult to comprehend. It seems that direct cell-cell contact is required and membranes are exchanged. Since Nef is associated with cellular membranes this might lead to some transfer of Nef to B cells. However, the immunological and functional consequences of this remain largely elusive. Alternatively, Nef-mediated manipulation of helper CD4 T cells might also impact B cell function and effective humoral immune responses. As previously noted, the presentation of the results and conclusions was in part very convoluted and difficult to comprehend. While the authors made attempts to improve the writing parts of the manuscript are still challenging to follow. This applies even more to the rebuttal (complex words combined with poor grammar), which made it difficult to assess which concerns have been satisfactory addressed.

We are grateful for the visionary comments. Based on suggestion, we have edited the writing throughout and appended remarks on certain points raised in the Discussion section. For points that need experimentation, we would like to address them in a follow-up study now under preparation.

**Reviewer #3 (Recommendations for the authors):**
Additional editing of the manuscript is highly recommended to make the results accessible for a broad readership.

We are grateful for the important suggestion. Accordingly, we have made editing of the manuscript aimed for a broad readership.